# Theory of Scaling Laws for In-Context Regression: Depth, Width, Context and Time

**Blake Bordelon**[♠], **Mary Letey**[◇♡], **Cengiz Pehlevan**[◇♡]
Center for Mathematical Sciences and Applications[♠]
John A. Paulson School of Engineering and Applied Sciences[◇]
Kempner Institute for the Study of Natural and Artificial Intelligence [♡],
Harvard University

## Abstract

We study in-context learning (ICL) of linear regression in a deep linear self-attention model, characterizing how performance depends on various computational and statistical resources (width, depth, number of training steps, batch size and data per context). In a joint limit where data dimension, context length, and residual stream width scale proportionally, we analyze the limiting asymptotics for three ICL settings: (1) isotropic covariates and tasks (ISO), (2) fixed and structured covariance (FS), and (3) where covariances are randomly rotated and structured (RRS). For ISO and FS settings, we find that depth only aids ICL performance if context length is limited. Alternatively, in the RRS setting where covariances change across contexts, increasing the depth leads to significant improvements in ICL, even at infinite context length. This provides a new solvable toy model of neural scaling laws which depends on both width and depth of a transformer and predicts an optimal transformer shape as a function of compute. This toy model enables computation of exact asymptotics for the risk as well as derivation of powerlaws under source/capacity conditions for the ICL tasks.

## 1 Introduction

In recent years, transformer models have become the backbone architecture for modern machine learning systems (Vaswani et al., 2017; Dosovitskiy et al., 2020). The transformer architecture consists of alternating self attention and multi-layer perceptron blocks in a deep residual network. For a variety of tasks, increasing the size of transformers by increasing width and depth of the model leads to empirically predictable improvements in model performance (Hestness et al., 2017; Kaplan et al., 2020; Hoffmann et al., 2022; Achiam et al., 2023). Various protocols for scaling up the width and depth of transformers have been developed that provide stable training limits (Yang et al., 2021; Bordelon et al., b; 2024b; Chizat & Netrapalli, 2024). One common strategy is to scale width $N$ and depth $L$ linearly with fixed aspect ratio $L/N$ (Noci et al., 2023; Dey et al., 2025), however existing theory cannot justify such a practice as compute optimal or distinguish relative performance gain from width and depth from total parameter count. This leads us to our first open question:

> **Q1: What sets optimal Transformer shapes and scaling laws?**
>
> How should width and depth scale under a compute budget in a transformer? Do transformer scaling laws only depend on total parameters or have different width and depth dependence?

We explore this for in-context learning (ICL) problems, specifically in-context linear regression problems. In-context learning refers to the ability of models to condition their outputs on the past sequence of inputs provided by the user *without updating model parameters* (Brown et al., 2020). This contrasts with in-weight learning (IWL) where information about the dataset is encoded in the model parameters during pre-training. Whereas prior theoretical works on neural scaling laws essentially analyze the role of width, pretraining data, or pretraining time (Bahri et al., 2021; Maloney et al., 2022; Simon et al., 2023; Bordelon et al., 2024a; Paquette et al., 2024; Lin et al., 2024; Bordelon et al., a), our work explores how depth, width, context length, and pretraining time influence the quality of ICL. We find that the architectural requirements to perform ICL regression depend significantly on the statistics of the pretraining ICL tasks. We are thus forced to investigate:

> **Q2: What task properties influence ICL solution**
>
> How does the statistical structure of ICL tasks influence the learned solution?

To address these questions, we develop a solvable model of deep linear attention for three distinct ICL data covariance structures. Our concrete novel contributions and findings are as follows:

## 1.1 CONTRIBUTIONS

1. We identify a new asymptotic scaling for in-context regression with linear attention where context length $P$, number of masked points per context $K$, and contexts per step $B$ all scale *linearly* with dimension $D$. This reduces the amount of compute and total data (by a factor of $D$) to converge than prior works (Lu et al., 2025) and speeds up simulations.

2. We introduce three distinct ICL data models of increasing generality. We study **isotropic** input data and task vectors (**ISO**). We then proceed to analyze fixed and structured (**FS**) covariances for data and task vectors. Lastly, we examine an ICL distribution where data covariances are randomly rotated and structured (**RRS**) across contexts.

3. We show with our asymptotic theory, that depth $L$ benefits the first two data settings (**ISO + FS**) only if $\alpha \equiv$ context length/data dimension is finite. In the limit of $\alpha \to \infty$, there is no benefit from depth for these ICL tasks. Further, the FS setting is brittle with respect to variations in changes to the data covariance at test time. However, for the **RRS** setting, increasing depth is beneficial for any distribution and a non-trivial width & depth scaling.

4. We introduce a model width $N$ bottleneck to our ICL model and study the pretraining scaling law for a model trained for $t$ steps on a width $N$ and depth $L$ linear attention model in the **RRS** setting. For powerlaw data, the scaling law takes the form of a Chinchilla scaling law but with width and depth contributing separate terms

$$\mathcal{L}(t, N, L, P) = c_t \, t^{-\beta_t} + c_N \, N^{-\beta_N} + c_L \, L^{-\beta_L} + c_P \, P^{-\beta_P}. \tag{1}$$

5. From this scaling law, we consider compute optimal joint scaling of width and depth. Depending on the structure of the ICL covariates, we obtain different scalings of $L \sim N^\nu$ where $\nu$ depends on properties of the data.

## 1.2 RELATED WORKS

**Infinite Neural Network Width and Depth Limits and Commonly Used Joint Scalings.** The empirical fact that larger networks tend to perform better on natural data (Hestness et al., 2017; Kaplan et al., 2020; Hoffmann et al., 2022) has led to development of scaling procedures to stably increase the size of a model to allow optimization. The mean-field or $\mu$P scaling theory (Geiger et al., 2020; Mei et al., 2018; Chizat & Bach, 2018; Yang & Hu, 2021; Bordelon & Pehlevan, 2022b) allows one to scale up the width of a model in a way that admits a feature learning infinite limit. Further, this scaling protocol provides consistent optimal hyperparameters, while delivering monotonic improvements in performance (Yang et al., 2021). The same program has been carried out for deep residual models, such as transformers (Bordelon et al., b; Yang et al., 2023; Bordelon et al., 2024b; Dey et al., 2025). However, while these infinite width and infinite depth scaling limits have been established to exist and perform better than finite models, no theory currently captures the *relative gains in performance from scaling up width or depth at fixed compute.* Understanding compute optimal shapes could help guide architectural choices when training large transformer models.

**Theories of Compute Optimal Neural Scaling Laws.** Following the empirical scaling law results of Kaplan et al. (2020) and Hoffmann et al. (2022), many theoretical works have examined the generalization theory for fully trained kernel methods under power law features (Bordelon et al., 2020; Spigler et al., 2020; Cui et al., 2021; Bahri et al., 2021; Maloney et al., 2022; Atanasov et al., 2023; 2022; Defilippis et al., 2024). More recently, several efforts have begun to incorporate SGD dynamics into these models to gain a notion of compute (and compute optimal tradeoffs between parameters and training time) (Bordelon & Pehlevan, 2022a; Bordelon et al., 2024a; Paquette et al., 2024; Lin et al., 2024; Kunstner & Bach, 2025; Bordelon et al., a). Lyu et al. (2025) recently extended scaling law theory to ICL by analyzing a single layer linear attention model performing ICL on a sparse multitask sparse feature regression problem, resulting in a neural scaling law in time and context length. In these works, the model is essentially one or two layers and the notion of finite model size is introduced with a random projection of the features to an $N$ dimensional space. In this way, existing theories more closely resemble scaling laws for width rather than a comparison

where depth and width serve different functions. Recent empirical works have pointed out the utility of increasing depth (or virtual depth through looping) for tasks requiring reasoning such as solving Sudoku puzzles (Wang et al., 2025) and solving math problems (Zhu et al., 2025; Geiping et al., 2025). Merrill & Sabharwal (2025) study regular language recognition where a clear computational advantage to scaling depth instead of width is established and experimentally verified.

**Empirical Studies of ICL.** Brown et al. (2020) demonstrated that large pretrained language models such as GPT-3 exhibit remarkable in-context learning capabilities for natural language tasks. Following this finding, Garg et al. (2022) empirically investigated the ability of transformers to learn simple function classes such as linear regression, sparse regression, and two-layer neural networks. Various works have studied emergence of ICL beyond language tasks, focusing on what data structures (from dynamical systems to classification problems) can be learned in-context (Liu et al., 2024b; Chan et al., 2022; He et al., 2024; McCracken et al., 2025; Park et al., 2024; Chen et al., 2024). Several works have offered experimental evidence that ICL implements Bayesian inference (Xie et al., 2021; Wurgaft et al., 2025; Liu et al., 2024a; Panwar et al., 2024; Zhang et al., 2023b). Further investigations questioned whether attention is even strictly necessary for ICL by studying the performance of MLPs on these tasks (Tong & Pehlevan, 2025; Kratsios & Furuya, 2025).

**Theoretical Studies of ICL.** Inspired by empirical studies of ICL, many works have theoretically investigated how ICL algorithms such as in-context gradient descent can be implemented in transformers (Zhang et al., 2023a; Ahn et al., 2023; Von Oswald et al., 2023; Li et al., 2023; Kim et al., 2024; Akyürek et al., 2022; Pathak et al., 2023; Mahankali et al., 2023). Some works have demonstrated flexibility of the transformer to adapt to changing statistics (Vladymyrov et al., 2024) or performing model selection (Bai et al., 2023). Studies have pointed out the need for sufficient pretraining task diversity for ICL generalization (Raventós et al., 2023; Wu et al., 2024) which was theoretically analyzed in an asymptotic scaling limit for a shallow linear attention model by Lu et al. (2025). Extensions to structured covariances and distribution shifts revealed the importance of train-test task alignment Letey et al. (2025). Gatmiry et al. (2024) investigate the need for a sufficient number of residual stream steps (either by increasing true depth or loops of attention layers) to solve ICL distributions with multiple condition numbers with larger condition numbers requiring more depth. While many of these works study a final construction of weights, recent theory has described the training dynamics of one layer linear multi-head attention (Zhang et al., 2025).

## 2 DATA, ARCHITECTURE AND REDUCED MODEL

**Deep Linear Attention Architecture** The most general model we study is a depth $L$, residual linear attention model $f$ that maps inputs contexts to output predictions. The data are formed as $P$ input-output pairs $\{(\boldsymbol{x}_\mu, y_\mu)\}_{\mu=1}^P$ and $K$ evaluation points $\{(\boldsymbol{x}_\mu^\star, *)\}_{\mu=P+1}^{P+K}$ (which do not carry target outputs) into a data matrix $\boldsymbol{D}$

$$\boldsymbol{D} = \begin{bmatrix} \boldsymbol{x}_1 & \dots & \boldsymbol{x}_P & \boldsymbol{x}_{P+1} & \dots & \boldsymbol{x}_{P+K} \\ y_1 & \dots & y_P & * & \dots & * \end{bmatrix} \tag{2}$$

where $*$ indicates masked target values on the $K$ evaluation points which are provided as $0$ entries. The evaluation tokens $\mu \in \{P+1, ..., P+K\}$ prevented from updating the model with a positional masking matrix $M_{\mu\nu}$ (Appendix B). The model $f_\mu$ is computed from

$$\boldsymbol{h}_\mu^1 = \boldsymbol{W}_x \boldsymbol{x}_\mu + \boldsymbol{w}_y y_\mu, \qquad \boldsymbol{h}_\mu^{\ell+1} = \boldsymbol{h}_\mu^\ell + \frac{1}{LP} \sum_{\nu=1}^P M_{\mu\nu} \left((\boldsymbol{k}_\nu^\ell)^\top \boldsymbol{q}_\mu^\ell\right) \boldsymbol{v}_\nu^\ell, \ \ell \in [L], \ \mu \in [P+K]$$

$$\boldsymbol{q}_\mu^\ell = \boldsymbol{W}_q^\ell \boldsymbol{h}_\mu^\ell, \ \boldsymbol{k}_\mu^\ell = \boldsymbol{W}_k^\ell \boldsymbol{h}_\mu^\ell, \ \boldsymbol{v}_\mu^\ell = \boldsymbol{W}_v^\ell \boldsymbol{h}_\mu^\ell, \qquad f_\mu = \boldsymbol{w}_o \cdot \boldsymbol{h}_\mu^L \tag{3}$$

where $\boldsymbol{W}_x \in \mathbb{R}^{N \times D}, \boldsymbol{w}_y, \boldsymbol{w}_o \in \mathbb{R}^N$ and $\boldsymbol{W}_j \in \mathbb{R}^{N \times N}$ for $j \in \{q, k, v\}$. The loss function for context $\boldsymbol{D}$ is $\mathcal{L}(\boldsymbol{D}) = \frac{1}{K} \sum_{\mu=P+1}^{P+K} (f_\mu - y_\mu)^2$ and the full population loss is $\mathcal{L} = \langle \mathcal{L}(\boldsymbol{D}) \rangle_{\boldsymbol{D}}$ where the average over $\boldsymbol{D}$ represents the distribution of context matrices. We stress that the operation $\boldsymbol{q}_\mu \cdot \boldsymbol{k}_\nu$ corresponds to *linear attention* rather than commonly used in soft-max attention. For the regression tasks we consider, this model is sufficient to solve the ICL task (Ahn et al., 2023; Von Oswald et al., 2023) and aids theoretical tractability (Lu et al., 2025; Zhang et al., 2025).

**Recurrent Reduced-$\Gamma$ Model** Following prior works on ICL in linear regression (Zhang et al., 2023a; Lu et al., 2025; Wu et al., 2024; Zhang et al., 2025), we examine the minimal (simplest) reparameterization of the above model which can solve this task where the residual stream encodes $\boldsymbol{x}$ information in a subspace orthogonal to $\boldsymbol{w}_y = \boldsymbol{w}_o$ so that $\boldsymbol{W}_x^\top \boldsymbol{w}_y = 0$ and $\boldsymbol{W}_v \propto \boldsymbol{w}_y \boldsymbol{w}_y^\top$. Instead of optimizing separate hidden weight matrices, we consider *looped / universal transformers* where

each attention block is identical $\boldsymbol{W}_j^\ell = \boldsymbol{W}_j^{\ell'}$ for $j \in \{k, q, v\}$ (Dehghani et al., 2018; Gatmiry et al., 2024). We relax this assumption in Section 5 and Appendix G.1 and show that untying layers does not alter the dynamics in our settings of interest. The reduced model defines the predictor $f(\boldsymbol{x}_\star)$ in terms of a single matrix $\boldsymbol{\Gamma} \in \mathbb{R}^{D \times D}$

$$\boldsymbol{\Gamma} \equiv \left( \boldsymbol{w}_o^\top \boldsymbol{W}_v \boldsymbol{w}_y \right) \boldsymbol{W}_x^\top \boldsymbol{W}_k^\top \boldsymbol{W}_q \boldsymbol{W}_x \ , \ f(\boldsymbol{x}_\star) = \frac{1}{LP} \boldsymbol{x}_\star^\top \boldsymbol{\Gamma} \sum_{\ell=0}^{L-1} \left( \boldsymbol{I} - L^{-1} \hat{\boldsymbol{\Sigma}} \boldsymbol{\Gamma} \right)^\ell \boldsymbol{X}^\top \boldsymbol{y}, \quad (4)$$

where $\hat{\boldsymbol{\Sigma}} = \frac{1}{P} \boldsymbol{X} \boldsymbol{X}^\top$ is the empirical covariance for the data matrix $\boldsymbol{X} \in \mathbb{R}^{D \times P}$ constructed from the first $P$ context vectors $\{\boldsymbol{x}_\mu\}_{\mu=1}^P$ with the first $P$ target values $\boldsymbol{y} \in \mathbb{R}^P$. Due to the simplicity of this model and loss landscape structure, we will first focus on the gradient dynamics when we directly perform gradient descent on the matrix $\boldsymbol{\Gamma}$. Then in Section 5 we consider learning dynamics for untied parameters across depth (non-recurrent models) and in Section 5.1 analyze gradient flow on decoupled attention blocks containing the full collection of parameters $\{\boldsymbol{W}_j\}_{j \in \{k, q, v\}}$.

# 3 LEARNING CURVES IN REDUCED LINEAR ATTENTION MODEL

## 3.1 ISOTROPIC COVARIATES AND TASKS (ISO)

To start our investigation, we begin by considering $D$ dimensional isotropic data and isotropic task distribution. For each context $\boldsymbol{D}_c$ and each data point $\mu \in [P]$, the distribution of $x$ and $y$ are

$$\boldsymbol{x}_{\mu,c} \sim \mathcal{N}(0, \boldsymbol{I}) \qquad \boldsymbol{\beta}_c \sim \mathcal{N}(0, \boldsymbol{I}) \ , \ y_{\mu c} = \frac{1}{\sqrt{D}} \boldsymbol{\beta}_c \cdot \boldsymbol{x}_c^\mu + \sigma \epsilon_c^\mu \ , \ \epsilon_c^\mu \sim \mathcal{N}(0, 1). \quad (5)$$

In Appendix C.1, we analyze SGD with batch of $B$ contexts sampled at each step in the proportional asymptotics $P, K, B, D \to \infty$ , $P/D = \alpha$ , $K/D = \kappa$ , $B/D = \tau$, establishing that successful pretraining requires a total of $Bt = \Theta(D)$ contexts, each of size $P = \Theta(D)$.

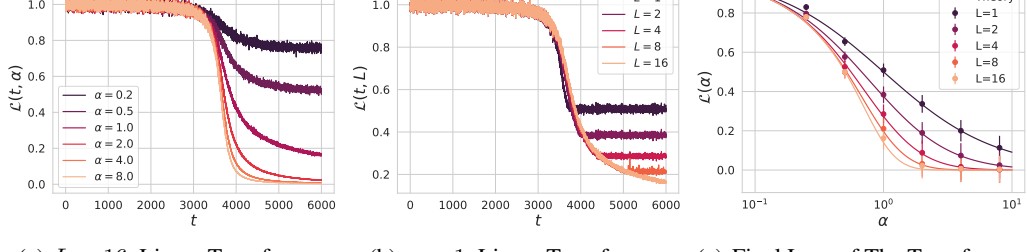

(a) $L = 16$, Linear Transformer    (b) $\alpha = 1$, Linear Transformer    (c) Final Loss of The Transformer

Figure 1: Deep linear self attention models trained with SGD on the ICL task with *isotropic covariates* with $D = 32$. (a) Training dynamics for varying $\alpha$. (b) Increasing depth $L$ can improve ICL predictions, especially for $\alpha \approx 1$. (c) The final loss compared to a theory of $L$ steps of GD.

These stochastic gradient descent fluctuation effects, while potentially interesting, are not the primary focus of the paper. Moving forward, we will instead focus on gradient flow in deeper models $L \geq 1$. For isotropic covariates, we have the following result.

---

**Result 1: Gradient Flow for Isotropic Covariates (ISO)**

Gradient flow ICL pretraining on isotropic data at any depth $L \geq 1$ from zero init yields

$$\boldsymbol{\Gamma}(t) = \gamma(t) \boldsymbol{I}, \quad (6)$$

where the dynamics obey a gradient flow for this scalar $\frac{d}{dt} \gamma(t) = -\partial_\gamma \mathcal{L}(\gamma, \alpha)|_{\gamma(t)}$ for

$$\mathcal{L}(\gamma, \alpha) = \int d\lambda \, \rho(\lambda, \alpha) \left[ \left( 1 - L^{-1} \gamma \lambda \right)^{2L} + \frac{\sigma^2}{\alpha \lambda} \left( 1 - \left[ 1 - L^{-1} \gamma \lambda \right]^L \right)^2 \right]. \quad (7)$$

The density $\rho(\lambda, \alpha)$ is the Marchenko-Pastur law with $\alpha \equiv P/D$.

---

The final loss $\mathcal{L}_\star(\alpha) = \min_\gamma \mathcal{L}(\gamma, \alpha)$ can be interpreted as the loss for $L$ steps of GD with optimal step size. We illustrate this effective loss function in Figure 2. Consider $\sigma^2 = 0$. For $L = 1$, the

loss saturates to $\mathcal{L}_\star = (1 + \alpha)^{-2}$ while for $L \to \infty$, the $\mathcal{L}_\star = [1 - \alpha]_+$, illustrating a gap in performance between shallow model and deep models at finite $\alpha$. Loss curves varying $L$ compared to linear transformers (dots) are shown in Figure 1 (c) and the $\mathcal{L}(\gamma)$ in Figure 2 (see App. C.2). For larger noise $\sigma^2$ (Figure 2 c), the optimal $\gamma$ is smaller since early stopping acts as an effective regularization (Advani et al., 2020; Sonthalia et al., 2024).

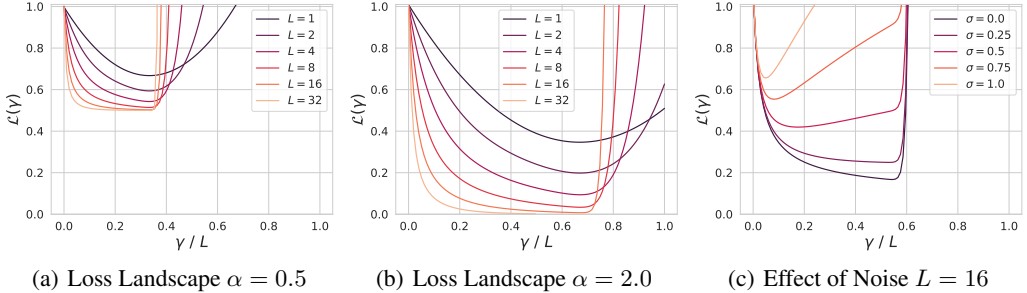

(a) Loss Landscape $\alpha = 0.5$    (b) Loss Landscape $\alpha = 2.0$    (c) Effect of Noise $L = 16$

Figure 2: The loss landscape for the reduced $\Gamma$ model with $\Gamma = \gamma I$ corresponding to the gradient flow limit. This limit is equivalent to **optimal step size** selection for in-context GD. (a)-(b) The effect of depth $L$ and context length $\alpha$ on the loss. (c) Larger noise $\sigma$ decreases the optimal $\gamma$.

> **Result 2: Depth is Unnecessary for Long Contexts on Isotropic Covariates**
>
> If $\alpha = P/D \to \infty$ then $L = 1$ achieves the minimal ICL loss (among any depth $L$ models). Any larger depth $L \geq 2$ achieves the same (zero if $\sigma^2 = 0$) loss in the $\alpha \to \infty$ limit.

### 3.2 Fixed Structured Covariance (FS) → Preconditioned Gradient Descent

Next, we let the population covariance be arbitrary $\langle xx^\top \rangle = \Sigma$ across all contexts and task correlations be given by the matrix $\langle \beta\beta^\top \rangle = \Omega$. The ICL population loss takes the form

$$\mathcal{L} = \text{tr}\left( \Omega \left\langle \left[ \left( I - L^{-1}\Gamma\hat{\Sigma} \right)^L \right]^\top \Sigma \left( I - L^{-1}\Gamma\hat{\Sigma} \right)^L \right\rangle \right) \tag{8}$$

> **Result 3: Depth Unnecessary for Long Contexts with Fixed Covariance $\Sigma$**
>
> When the ICL distribution involves fixed covariance across contexts, there is no benefit to increasing depth $L$ beyond $L = 1$ in the large context $\alpha \to \infty$ limit. For any $L \geq 1$, zero ICL loss can be achieved in the $\alpha \to \infty$ limit by setting $\Gamma = L\Sigma^{-1}$.

We support this finding in Figure 3 (b) where we show that small depth models are not outperformed by deeper models even after very long training horizons. When the ICL pretraining distribution for $\Sigma$ is fixed, the model will memorize statistical information about the covariance of the inputs from the pretraining distribution. By preconditioning with the inverse of the data covariance $\Sigma^{-1}$, the model is capable of achieving zero loss after even a single step of in-context GD. The gradient flow dynamics of the $\Gamma$ matrix can be decomposed further in the case where $\Omega$ and $\Sigma$ commute.

> **Result 4: Decoupled Eigenvalue Dynamics During Pretraining on Fixed Covariances**
>
> Suppose that $\Omega$ and $\Sigma$ are codiagonalizable with respective eigenvalues $\{\omega_k\}$ and $\{\lambda_k\}$. Then, when training from zero initialization, $\Gamma$ is diagonal in the same basis with eigenvalues $\gamma_k(t)$ that obey the following dynamics (see Appendix D)
>
> $$\frac{d}{dt}\gamma_k(t) = \omega_k \lambda_k^2 \left( 1 - L^{-1}\lambda_k\gamma_k(t) \right)^{2L-1}. \tag{9}$$
>
> For $L \to \infty$ these dynamics have solution $\gamma_k(t) = \frac{1}{2\lambda_k}\ln(1 + 4\omega_k\lambda_k^3 t)$ generating the loss dynamics $\lim_{L\to\infty} \mathcal{L}(t, L) = \sum_k \frac{\omega_k\lambda_k}{1+4\omega_k\lambda_k^3 t}$. Under powerlaw conditions $\lambda_k \sim k^{-\nu}$ and $\sum_{\ell > k} \lambda_\ell\omega_\ell \sim k^{-\nu\beta}$, the ICL loss scales as a powerlaw $\mathcal{L}(t) \sim t^{-\frac{\beta}{\nu+\nu\beta+1}}$.

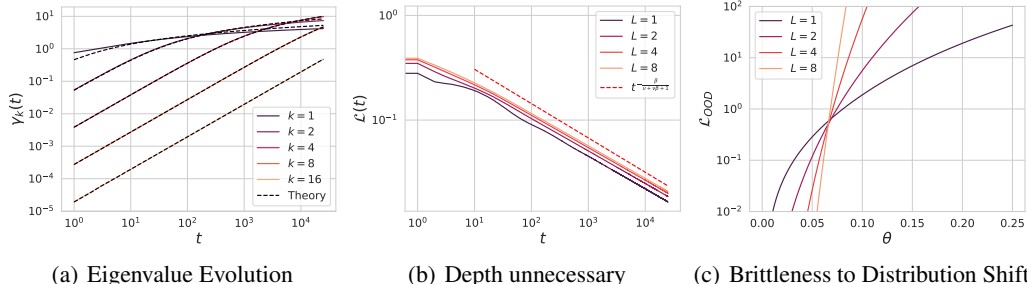

(a) Eigenvalue Evolution  (b) Depth unnecessary  (c) Brittleness to Distribution Shift

Figure 3: Pretraining on **FS** ICL covariates leads to a solution that does not require depth but is brittle to distribution shift. (a) Evolution of the eigenvalues $\gamma_k(t)$ of the $\mathbf{\Gamma}(t)$ matrix for depth $L = 4$ as a function of pretraining time $t$ compared with infinite depth $L \to \infty$ theory (dashed black). (b) For powerlaw covariates, all depth models converge as a power law in $t$. There is no asymptotic benefit to increasing depth beyond $L = 1$. (c) The ICL solution obtained when training from fixed covariance is **brittle** to changes in the covariance $\mathbf{\Sigma} \to \exp(-\theta\mathbf{S})\mathbf{\Sigma}\exp(\theta\mathbf{S})$.

While pretraining the linear transformer in this setting can achieve zero loss for long contexts $P \to \infty$ even in shallow networks, the solution the model finds from gradient descent is *brittle* to properties of the pretraining data. Rather than learning a generic algorithm that solves ICL regression for any covariance $\mathbf{\Sigma}$, its solution is specialized to the pretraining covariance.

---

**Result 5: Brittleness of Fully Trained Model to Distribution Shift (OOD Loss)**

A depth $L$ model is pretrained on ICL tasks with fixed covariances $\mathbf{\Sigma}$ and $\mathbf{\Omega}$ for inputs and task vectors, but evaluated on new covariances $\mathbf{\Sigma}', \mathbf{\Omega}'$. The out-of-distribution loss is

$$\mathcal{L}_{\text{OOD}} = \text{tr}\left(\mathbf{\Omega}'\left[\left(\mathbf{I} - \mathbf{\Sigma}^{-1}\mathbf{\Sigma}'\right)^L\right]^\top \mathbf{\Sigma}'\left(\mathbf{I} - \mathbf{\Sigma}^{-1}\mathbf{\Sigma}'\right)^L\right) \tag{10}$$

---

We illustrate this brittleness in Figure 3 (c) where we define a family of new covariance matrices $\mathbf{\Sigma}' = \exp(\theta\mathbf{S})\mathbf{\Sigma}\exp(-\theta\mathbf{S})$ where $\mathbf{S}$ is a random skew-symmetric matrix. As $\theta$ increases and $\mathbf{\Sigma}'$ becomes more dissimilar to $\mathbf{\Sigma}$, the OOD loss increases for all depths $L$.

## 3.3 Random Rotated and Structured Covariances $\implies$ In-Context GD

Next, we attempt to enhance the *covariance diversity* across contexts. To do so, we now allow that each context $c$ has a random data and task covariances which are randomly rotated

$$\boldsymbol{x}_c^\mu \sim \mathcal{N}\left(0, \mathbf{\Sigma}_c\right)\ ,\ \mathbf{\Sigma}_c = \boldsymbol{O}_c\mathbf{\Lambda}\boldsymbol{O}_c^\top\ ,\ \mathbf{\Omega}_c = \boldsymbol{O}_c\mathbf{\Omega}\boldsymbol{O}_c^\top, \tag{11}$$

where $\boldsymbol{O}_c$ is a random $d \times d$ orthogonal matrix sampled from the Haar measure. The idea to pretrain with a diverse set of covariances $\mathbf{\Sigma}_c$ across contexts $c$ is to encourage the model to learn a *generic* in-context learning algorithm that is not specifically tailored to a particular data covariance $\mathbf{\Sigma}$. By introducing the random rotation across contexts, the model cannot encode a whitening transform of the data in the matrix $\mathbf{\Gamma}$ which prevents a zero loss solution in a shallow model with depth $L = 1$. Therefore, even the $P/D \to \infty$ limit has the potential to exhibit a nontrivial scaling law in $L$.

---

**Result 6: Gradient Flow Generates Isotropic $\mathbf{\Gamma}$**

Gradient flow on the $\mathbf{\Gamma}$-reduced model maintains the isotropy condition $\mathbf{\Gamma}(t) = \gamma(t)\boldsymbol{I}$ with

$$\frac{d}{dt}\gamma(t) = \text{tr}\left(\mathbf{\Lambda}^2\mathbf{\Omega}\left(\boldsymbol{I} - L^{-1}\gamma(t)\mathbf{\Lambda}\right)^{2L-1}\right). \tag{12}$$

---

This indicates that, provided the covariance is randomly rotated across contexts, the behavior of the learned solution is unconditioned in-context GD (see Appendix E). In the next section, we explore the consequences of this finding for optimal shapes.

## 4 MODEL OF COMPUTE OPTIMAL NEURAL SCALING LAWS

We consider the third setting with power law features and also introduce a notion of width through a projection matrix $\boldsymbol{A} \in \mathbb{R}^{N \times D}$. Rather than training on $D$ dimensional inputs $\boldsymbol{x}$, the model has access to $N$ dimensional features $\tilde{\boldsymbol{x}} = \boldsymbol{A}\boldsymbol{x}$, which leads to the following condition of $\boldsymbol{\Gamma}(t)$

$$\tilde{\boldsymbol{x}} = \boldsymbol{A}\boldsymbol{x} \implies \boldsymbol{\Gamma}(t) = \gamma(t)\left(\boldsymbol{A}\boldsymbol{A}^{\top}\right) \in \mathbb{R}^{N \times N} \tag{13}$$

In this section, we consider *arbitrary* (non-proportional) but large values of $N, P, D$, which requires an approximate non-proportional mean field theory (see Appendix F.1). This theory allows for the possibility of trace class covariates $\sum_{k=1}^{\infty} \lambda_k < \infty$ for $D \to \infty$. As before, the loss can again be viewed as a function of the scale parameter $\gamma(t)$. We provide a recipe to compute it below by computing the average over the random orthogonal matrix, resulting in a two-point deterministic equivalent for free products [1], in the same spirit of the results of (Bordelon et al., 2024a; Atanasov et al., 2025), using a saddle point method (Appendix A, F.1). As the matrix $\boldsymbol{M} = \boldsymbol{O}(\boldsymbol{A}^{\top}\boldsymbol{A})^2\boldsymbol{O}^{\top}\hat{\boldsymbol{\Sigma}}$ driving the dynamics is asymmetric, we characterize the loss landscape using dynamical mean field theory, a technique from the physics of spin glasses which allows asymptotic descriptions of high dimensional disordered dynamical systems (Sompolinsky & Zippelius, 1981).

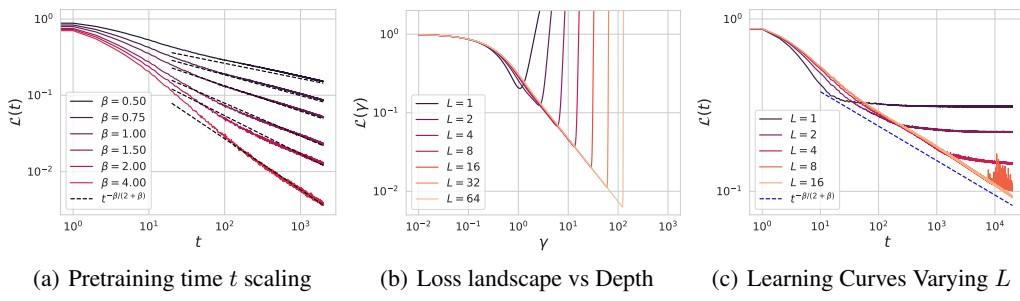

(a) Pretraining time $t$ scaling     (b) Loss landscape vs Depth     (c) Learning Curves Varying $L$

Figure 4: Loss dynamics for powerlaw **RRS** data. (a) Varying the source exponent $\beta$, we see that the scaling with pretraining time has exponent $\frac{\beta}{2+\beta}$. (b) The loss landscape across depths $L$ for the scalar $\gamma$ parameter exhibits minima at $\gamma \approx L$. (c) The training dynamics of the reduced-$\Gamma$ model exhibit $t^{-\beta/(2+\beta)}$ decay before hitting an asymptote which scales as $L^{-\beta}$.

---

**Result 7: Two-Point Deterministic Equivalent (DMFT Correlation) for the Loss Landscape**

The loss function $\mathcal{L} = \left\langle |\boldsymbol{\Lambda}^{1/2}[\boldsymbol{I} - \gamma(t)L^{-1}\boldsymbol{O}\left(\boldsymbol{A}^{\top}\boldsymbol{A}\right)^2\boldsymbol{O}^{\top}\hat{\boldsymbol{\Sigma}}]^L \bar{\boldsymbol{\beta}}|^2 \right\rangle$ can be explicitly averaged over the random orthogonal matrix $\boldsymbol{O}$ and expressed as a deterministic function

$$\mathcal{L}(t, N, L, P) = \int \frac{d\omega \, d\omega'}{(2\pi)^2}\left(1 + L^{-1}\gamma(t)i\omega\right)^L \left(1 + L^{-1}\gamma(t)i\omega'\right)^L \mathcal{C}(\omega, \omega')$$

$$\mathcal{C}(\omega, \omega') \simeq \mathrm{Tr}\left(\boldsymbol{\Lambda}\left[i\omega\boldsymbol{I} + \Psi_{v\chi}(\omega)\boldsymbol{\Lambda}\right]^{-1}\left[\boldsymbol{\Omega} - \Psi_{\chi\chi}(\omega, \omega')\boldsymbol{I}\right]\left[i\omega'\boldsymbol{I} + \Psi_{v\chi}(\omega')\boldsymbol{\Lambda}\right]^{-1}\right)$$

$$\tag{14}$$

where $\Psi_{v\chi}(\omega)$ and $\Psi_{\chi\chi}(\omega, \omega')$ are deterministic functions that depend on the spectra of $\hat{\boldsymbol{\Sigma}}$ and $\boldsymbol{A}^{\top}\boldsymbol{A}$ (Appendix A, F.1). For example to obtain $\Psi_{v\chi}(\omega)$ we first solve

$$i\omega_{(\boldsymbol{A}^{\top}\boldsymbol{A})^2}(\tau)i\omega_{\hat{\boldsymbol{\Sigma}}}(\tau) = \left(-1 + \frac{D}{\tau}\right)i\omega \implies \Psi_{v\chi}(\omega) = \frac{i\omega}{i\omega_{(\boldsymbol{A}^{\top}\boldsymbol{A})^2}(\tau)} \tag{15}$$

where $\tau$ and $\omega_M$ for matrix $M$ is defined as $\tau = \mathrm{Tr}\, \boldsymbol{M}\left(\boldsymbol{M} + i\omega_M\right)^{-1}$.

---

In Appendix F.5 we provide formulas for $\Psi_{v\chi}(\omega)$ and $\Psi_{vv}(\omega, \omega')$ in the case that $\boldsymbol{A}^{\top}\boldsymbol{A} \in \mathbb{R}^{D \times D}$ is a rank $N$ projection (has $N$ eigenvalues equal to 1) and $\hat{\boldsymbol{\Sigma}} = \frac{1}{P}\boldsymbol{X}\boldsymbol{X}^{\top} \in \mathbb{R}^{D \times D}$ is a structured

---

[1]*Two-point* refers to the correlation function of two resolvents evaluated at different arguments, rather than the one point function which is a single resolvent and determines only the spectrum of the random matrix.

Wishart matrix. We can further extract Chinchilla scaling laws for powerlaw data. We demonstrate the need for both width and depth in Figures 4 and 5. We verify the scaling exponents for time and depth in Figure 4 and exhibit the compute optimal scaling law for width and depth in Figure 5.

---

**Result 8: Scaling Law for Power Law Features and Optimal Shapes**

Assume source and capacity conditions for the eigenvalues and target coefficients $\omega_k \equiv \Omega_{kk}$

$$\sum_{\ell > k} \lambda_\ell \, \omega_k \sim k^{-\nu\beta} \; , \; \lambda_k \sim k^{-\nu}, \tag{16}$$

with source/capacity exponents $(\beta, \nu)$. Let $\boldsymbol{A}$ be rank $N$ and frozen. Then the loss follows a neural scaling law in the resources of time $t$, width $N$, depth $L$, and context length $P$

$$\mathcal{L}(t, N, L, P) \approx c_t \, t^{-\frac{\beta}{2+\beta}} + c_N \, N^{-\nu\beta} + c_L \, L^{-\beta} + c_P \, P^{-\nu\beta}. \tag{17}$$

As a consequence, at fixed compute $C = tP^2N^2L$ the optimal width and depth scale as $L \propto N^\nu$.

---

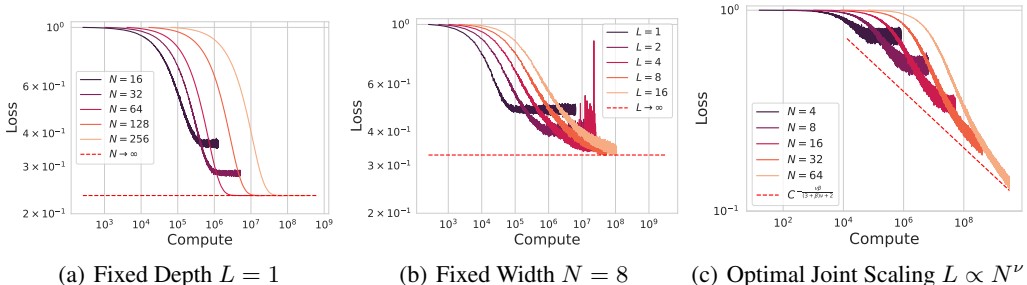

(a) Fixed Depth $L = 1$      (b) Fixed Width $N = 8$      (c) Optimal Joint Scaling $L \propto N^\nu$

Figure 5: Increasing width and depth alone is insufficient to obtain monotonic improvements on powerlaw data with random covariance across contexts. (a) Scaling only width leads to a depth bottleneck (dashed red line). (b) Scaling only depth leads to a width bottleneck (dashed red line). (c) Increasing $N$ and $L$ simultaneously achieves monotonic improvement with compute.

## 5   MORE REALISTIC SELF ATTENTION MODELS

In this section, we discuss more realistic models which exhibit similar training dynamics and depth dependence. First, we describe the dynamics when the $\boldsymbol{\Gamma}^\ell$ matrices are not tied. Second, we provide a theory of training when each of the weight matrices is optimized separately. Lastly, we provide experiments with softmax attention models trained with Adam.

**Gradient Flow with Untied $\Gamma$ Matrices**   In this section, we consider the effect of untied $\boldsymbol{\Gamma}^\ell$ matrices across layers $\ell \in [L]$. When each of the weights are optimized separately with a learning rate that is upscaled by depth $\eta = \eta_0 L$, the dynamics are actually equivalent to the recurrent model that we presented previously under the **RRS** noise-free setting.

---

**Result 9: Gradient Flow with Untied Layers on RRS Covariates**

Let the model prediction on evaluation point $\boldsymbol{x}_\star$ for dataset $\boldsymbol{D}$ after gradient flow time $t$ for pretraining on **RRS** covariates with $L$ decoupled matrices $\{\boldsymbol{\Gamma}^\ell\}_{\ell=1}^L$ and learning rate $\eta = \eta_0 L$ be written as $f_{\text{untied}}(\boldsymbol{x}_\star, t)$. Let the predictor resulting from gradient flow on tied matrices $\boldsymbol{\Gamma}^\ell = \boldsymbol{\Gamma}$ (looped transformer) with learning rate $\eta_0$ be $f_{\text{tied}}(\boldsymbol{x}_\star, t)$. If the initial condition for the matrices $\boldsymbol{\Gamma}^\ell|_{t=0} = \gamma(0)\boldsymbol{I}$ are isotropic and equal for all $\ell$, then

$$f_{\text{untied}}(\boldsymbol{x}_\star, t) = f_{\text{tied}}(\boldsymbol{x}_\star, t) \;\; \forall t \geq 0, \tag{18}$$

and consequently the ICL losses are also equivalent through time $\mathcal{L}_{\text{untied}}(t) = \mathcal{L}_{\text{tied}}(t)$.

---

We prove this result in Appendix G.2, which relies on the permutation symmetry in the predictor function $f$ with respect to the variables $\{\gamma^\ell\}_{\ell=1}^L$. Since under RRS covariates, the $\mathbf{\Gamma}^\ell$ matrices remain isotropic through all gradient flow training $\mathbf{\Gamma}^\ell(t) = \gamma^\ell(t)\mathbf{I}$, all the matrices arising in the predictor dynamics commute, resulting in the permutation symmetry.

In Figure 8 we provide a numerical demonstration that under equal identical conditions (such as $\mathbf{\Gamma}^\ell = 0$ for all $\ell$), that the $\gamma^\ell$ scalars remain equal for all training times $t$ when pretraining on RRS covariates. However, we also provide evidence that this symmetry requires a symmetric initial condition. When starting from an asymmetric initial condition $\gamma^\ell \neq \gamma^k$, the model does not necessarily converge to a symmetric configuration.

## 5.1 Gradient Flow for Full Linear Attention

In this section we consider gradient flow on all of the attention weights $\{\mathbf{W}_k, \mathbf{W}_q, \mathbf{W}_v\}$ separately rather than gradient flow on the $\mathbf{\Gamma}$ matrix. This corresponds to the dynamical system

$$\frac{d}{dt}\mathbf{W}_j = -\frac{\partial}{\partial \mathbf{W}_j}\mathcal{L} \, , \, j \in \{x, y, k, q, v, o\} \tag{19}$$

The dynamics of this model from small initialization are theoretically tractable as a set of low dimensional ODEs (see Appendix G.2), but suffers some defects due to transient blowup and recovery in the scale of $\mathbf{w}_y$ and $\mathbf{w}_o$. However, if we fix these weights to unit norm, the dynamics of the above model reduces to a *one-dimensional ODE* much like the reduced-$\mathbf{\Gamma}$ model.

---

**Result 10: Gradient Flow on Full Linear Attention Module**

Consider pretraining a linear transformer on randomly rotated covariance ICL data distribution. Fix read-in weights $\mathbf{w}_y$ and readout weights $\mathbf{w}_o$ with $\mathbf{w}_y = \mathbf{w}_o$ and $|\mathbf{w}_y| = 1$. Initialize the other weights to be small $\frac{1}{2}|\mathbf{W}_x|^2 = |\mathbf{W}_k|^2 = |\mathbf{W}_q|^2 = |\mathbf{W}_v|^2 = \sigma^2$ where $\sigma \ll 1$. The gradient dynamics will maintain a balanced condition where $|\mathbf{W}_j(t)| = |\mathbf{W}_{j'}(t)| = w(t)$ for $j, j' \in \{x, k, q, v\}$ where the scalar $w(t)$ evolves as

$$\frac{d}{dt}w(t) = w(t)^4 \operatorname{tr}\left(\mathbf{\Lambda}^2 \mathbf{\Omega}\left(\mathbf{I} - w(t)^5\mathbf{\Lambda}\right)^{2L-1}\right) \tag{20}$$

This can be interpreted as gradient flow on the loss function for the reduced $\mathbf{\Gamma}$ model under the reparameterization $\gamma(t) \to w(t)^5$. For powerlaw covariates with source exponent $\beta$, this gives a powerlaw scaling with pretraining time $t$ and depth $L$

$$\mathcal{L}(t, L) \sim c_t \, t^{-\frac{5\beta}{5\beta+2}} + c_L \, L^{-\beta}. \tag{21}$$

---

We show the learning curves for this full attention module for isotropic covariates and powerlaw covariates in Figure 6. The theoretical dynamics and predicted powerlaw exponent for this new dynamical system closely match the predictions.

## 5.2 Nonlinear (Softmax) Attention, Multiple Heads, MLPs

We also provide experiments showing that similar scaling behavior holds in softmax attention models trained with Adam. In this setting, the attention block has the form $\mathbf{h}_\mu^{\ell+1} = \mathbf{h}_\mu^\ell + L^{-1}\sum_{\nu=1}^P A_{\mu\nu}\mathbf{W}_v\mathbf{h}_\nu^\ell$ where $A_{\mu\nu}^\ell = \frac{1}{Z_\mu}\exp(\mathbf{h}_\nu^\top\mathbf{W}_k\mathbf{W}_q\mathbf{h}_\mu^\ell)$ with $Z_\mu = \sum_\nu \exp(\mathbf{h}_\nu^\top\mathbf{W}_k\mathbf{W}_q\mathbf{h}_\mu^\ell)$ as the normalization factor. We train with Adam and show depth is still beneficial to ICL performance in Figure 6 (c) and 9, consistent with the phenomenology of our linear attention model. We also provide additional experiments varying heads, introducing MLP blocks in softmax attention models in Figure 10.

## 6 Discussion

**Conclusion** In this work we analyzed a solvable model of in-context learning with deep linear attention. We showed that the pretraining statistics strongly determine the type of solution the model converges to and under what conditions scaling depth is necessary at large context lengths. When training on fixed covariance structures, large depth is not necessary (at infinite context length) as the model learns a preconditioner that whitens the data. However, this learned solution is brittle to changes in the data covariance. When training on tasks where the data covariance is randomly

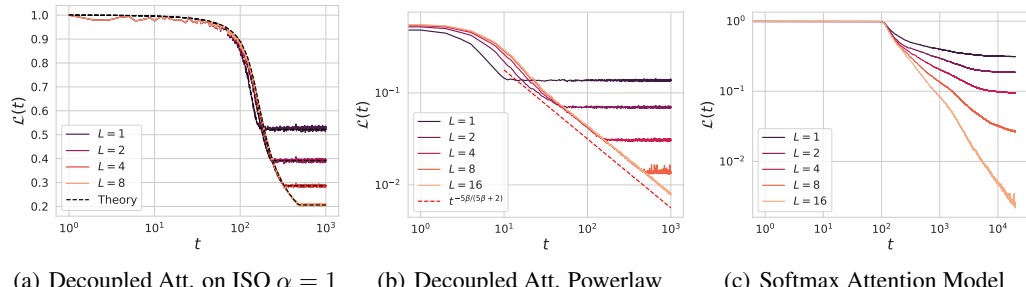

(a) Decoupled Att. on ISO $\alpha = 1$     (b) Decoupled Att. Powerlaw     (c) Softmax Attention Model

Figure 6: (a)-(b) Dynamics of the full decoupled linear attention (decoupled att.) module with separate $\{W_x, W_q, W_k, W_v\}$ as a function of depth $L$ agree with a theory generated through reparameterization $\gamma \to w^5$ as described in Result 10. (c) Softmax attention dynamics across depths $L$ indicate similar improvements to performance from depth. We provide additional experiments varying attention heads, depth, and the addition of MLP blocks in Figure 9 and 10.

rotated across contexts, the model learns a general purpose in-context gradient descent algorithm and exhibits a separable Chinchilla neural scaling law where limited width and depth can both bottleneck performance. Lastly, we show these results are robust to reparameterization of the attention blocks.

**Limitations and Future Directions** While our work presents an advance in the solvable models of neural scaling laws and the structure of ICL in linear attention, there are many current limitations. The primary limitation is that we focus on linear regression tasks solved with linear attention. Characterization of more general ICL problems such as nonlinear function approximation and nonlinear attention could provide more insights into realistic in-context function approximation. Further, our analysis is focused on online learning in the present work. Future work could investigate overfitting effects caused by repeating tasks or context matrices during training, perhaps with dynamical mean field theory techniques (Mignacco et al., 2020; Bordelon et al., 2024a; Montanari & Urbani, 2025). While this current work identifies heterogeneity in data covariances as necessary for self-attention models to discover ICL solutions that utilize depth to generalize across covariances, future work could explore diverse covariances beyond the RRS construction that our work focuses on including data domain shift, changes in label noise level, and hierarchical data structure. Future work could also examine the role of large learning rate effects during pretraining dynamics. Another direction that could be interesting to explore in future works is scaling up the number of loops dynamically as the network trains to significantly reduce the total compute required.

ACKNOWLEDGEMENTS

The authors would like to thank Jacob Zavatone-Veth, Alex Atanasov, Alexandru Meterez, Clarissa Lauditi, William Tong, Jamie Simon, Boris Hanin, Zhouran Yang, and Jascha Sohl-Dickstein for insightful discussions. B.B. acknowledges support from the Center of Mathematical Sciences and Applications(CMSA) of Harvard University. C.P. is supported by an NSF CAREER Award (IIS-2239780), DARPA grants DIAL-FP-038 and AIQ-HR00112520041, the Simons Collaboration on the Physics of Learning and Neural Computation, and the William F. Milton Fund from Harvard University. This work has been made possible in part by a gift from the Chan Zuckerberg Initiative Foundation to establish the Kempner Institute for the Study of Natural and Artificial Intelligence.

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

APPENDIX

## A  TWO POINT DETERMINISTIC EQUIVALENT FOR GENERAL FREE PRODUCT FROM DYNAMICAL MEAN FIELD THEORY

In this section, we describe a method to handle two-point correlation properties of free products of random matrices. Similar results were provided in a simple random feature model of (Bordelon et al., 2024a; Atanasov et al., 2025). This computation provides the necessary technical results to establish the behavior of ICL in the randomly rotated context setting. Note throughout that we will use $\mathrm{tr} M$ for normalized trace of the matrix $M$ so that for $N \times N$ matrix $\mathrm{tr} M = \frac{1}{N} \sum_{k=1}^{N} M_{kk}$. We instead study arbitrary (possibly trace class) covariates in Appendix F.1.

**Intuitive Motivation for this Section**    Before working out this random matrix result, we first want to motivate it. To do so, we start by considering the infinite depth limit $L \to \infty$ of the model in Section 4 before describing how our results will also cover finite depth $L$. In the infinite depth $L \to \infty$ limit, the loss can be expressed as the as a dynamical system in layer-time $\tau$ (not gradient descent time) evaluated at $\tau = \gamma$

$$\mathcal{L}(\gamma) = \mathrm{tr} \mathbf{\Lambda} \left\langle \boldsymbol{v}(\gamma) \boldsymbol{v}(\gamma)^\top \right\rangle \ , \ \partial_\tau \boldsymbol{v}(\tau) = -\boldsymbol{M} \boldsymbol{v}(\tau) + \delta(\tau) \boldsymbol{\beta}. \tag{22}$$

where the average $\langle \rangle$ is over both the matrix $\boldsymbol{M}$ and the initial condition $\boldsymbol{\beta}$ which has correlation $\left\langle \boldsymbol{\beta} \boldsymbol{\beta}^\top \right\rangle = \mathbf{\Omega}$. We note that for the RRS setting, the matrix $\boldsymbol{M}$ is generally an *asymmetric* free product of the form $\boldsymbol{M} = \boldsymbol{O} \boldsymbol{B} \boldsymbol{O}^\top \boldsymbol{A}$ where $\boldsymbol{B}$ and $\boldsymbol{A}$ have known spectrum. A key idea is to examine the Fourier transform of the correlation matrix for $\boldsymbol{v}(\tau)$

$$\left\langle \boldsymbol{v}(\gamma) \boldsymbol{v}(\gamma)^\top \right\rangle = \int \frac{d\omega d\omega'}{(2\pi)^2} e^{i\gamma(\omega + \omega')} \left\langle [i\omega + \boldsymbol{M}]^{-1} \mathbf{\Omega} \left[ i\omega' + \boldsymbol{M}^\top \right]^{-1} \right\rangle \tag{23}$$

We therefore see that the loss depends on a two-point interaction of the resolvent matrices $[i\omega + \boldsymbol{M}]^{-1}$ evaluated at different frequencies $\omega, \omega'$. For finite depth $L$ the result can again be expressed in terms of this two-point resolvent correlation using Cauchy's Integral formula

$$\left\langle \boldsymbol{v}^L (\boldsymbol{v}^L)^\top \right\rangle = \int \frac{d\omega d\omega'}{(2\pi)^2} (1 - \gamma L^{-1} i\omega)^L (1 - \gamma L^{-1} i\omega')^L \left\langle [i\omega + \boldsymbol{M}]^{-1} \mathbf{\Omega} \left[ i\omega' + \boldsymbol{M}^\top \right]^{-1} \right\rangle. \tag{24}$$

Thus for any depth $L$ the ICL loss is governed by the correlation function in frequency space

$$\boldsymbol{C}(\omega, \omega') = \left\langle \boldsymbol{v}(\omega) \boldsymbol{v}(\omega')^\top \right\rangle = \left\langle [i\omega + \boldsymbol{M}]^{-1} \mathbf{\Omega} \left[ i\omega' + \boldsymbol{M}^\top \right]^{-1} \right\rangle \tag{25}$$

is a valuable summary statistic to compute the test loss when averaging over the RRS model's random rotation matrix $\boldsymbol{O}$. We can therefore consider a simple linear dynamical system in continuous time and taking a Fourier transform to obtain $\boldsymbol{C}(\omega, \omega')$.

To compute the pretraining dynamics for the general random $\boldsymbol{M}$, one must compute gradients of the effective loss with respect to $\gamma$. For example, at infinite depth, the gradient flow pretraining dynamics are

$$\frac{d}{dt} \gamma(t) = - \int \frac{d\omega d\omega'}{(2\pi)^2} i(\omega + \omega') e^{i\gamma(\omega + \omega')} \mathrm{tr} \, \mathbf{\Lambda} \, \boldsymbol{C}(\omega, \omega'). \tag{26}$$

Now that we have motivated computing the $\boldsymbol{C}(\omega, \omega')$ object, we will now give a very simple Gaussian example before deriving the general free product result necessary for the RRS covariates.

**Dynamical Mean Field Theory Illustrative Example**    To illustrate what Dynamical Mean Field Theory computes, we will start by deriving a deterministic equivalent result for a simple $N \times N$ Gaussian Orthogonal Ensemble (GOE) matrix $M_{ij} = M_{ji} \sim \mathcal{N}(0, 1/N)$ [2].

$$\frac{d}{dt} \boldsymbol{v}(t) = -\boldsymbol{M} \boldsymbol{v}(t) + \delta(t) \boldsymbol{v}_0. \tag{27}$$

---

[2] Because this matrix is symmetric the two-point equivalent is not necessary since $\exp(-\boldsymbol{M} t) \exp(-\boldsymbol{M}^\top t) = \exp(-2\boldsymbol{M} t)$ allowing for a Fourier transform in a single time variable. We still compute the two point function as an illustrative example.

As $N \to \infty$, this dynamical system can be replaced with an *uncoupled* dynamical system which depends on some summary variables known as correlation $C(t, t')$ and response function $R(\tau)$

$$\frac{d}{dt}\boldsymbol{v}(t) = \boldsymbol{\xi}(t) + \int dt' R(t-t')\boldsymbol{v}(t') + \delta(t)\boldsymbol{v}_0 \tag{28}$$

$$\left\langle \boldsymbol{\xi}(t)\boldsymbol{\xi}(t')^{\top} \right\rangle = C(t,t')\boldsymbol{I} \ , \ C(t,t') = \text{tr}\left\langle \boldsymbol{v}(t)\boldsymbol{v}(t')^{\top} \right\rangle \ , \ R(\tau) = \text{tr}\left\langle \frac{\partial \boldsymbol{v}(t+\tau)}{\partial \boldsymbol{\xi}(t)^{\top}} \right\rangle \tag{29}$$

Taking a Fourier transform of the above equation $\boldsymbol{v}(\omega) = \int dt e^{-i\omega t}\boldsymbol{v}(t)$, we find

$$\boldsymbol{v}(\omega) = [i\omega + R(\omega)]^{-1}[\boldsymbol{v}_0 + \boldsymbol{\xi}(\omega)] \tag{30}$$

where $R(\omega)$ is the Fourier transform of the response function. This function

$$i\omega R(\omega) = 1 + R(\omega)^2 \tag{31}$$

The correlation function $\boldsymbol{C}(\omega, \omega') = \left\langle \boldsymbol{v}(\omega)\boldsymbol{v}(\omega')^{\top} \right\rangle$ thus satisfies the equation

$$\boldsymbol{C}(\omega, \omega') = [i\omega + R(\omega)]^{-1}[i\omega' + R(\omega')]^{-1}\left[\left\langle \boldsymbol{v}_0\boldsymbol{v}_0^{\top} \right\rangle + \boldsymbol{C}(\omega, \omega')\right]. \tag{32}$$

We see that the DMFT description enabled an average over the random matrix to compute $\boldsymbol{C}(\omega, \omega')$. This result is exact as the matrix size $N \to \infty$. We would now like to perform a similar analysis for a general free product matrix (to deal with our RRS ensemble) which involves averaging over orthogonal rather than Gaussian matrices.

**Tracking Linear Dynamics Generated by Free Product** Taking this dynamical approach, we consider the following dynamical system which depends on a random matrix $\boldsymbol{M} \in \mathbb{R}^{N \times N}$

$$\partial_t \boldsymbol{v}(t) = -\boldsymbol{M}\boldsymbol{v}(t) + \delta(t)\boldsymbol{v}_0 \ , \ \boldsymbol{M} = \boldsymbol{O}\boldsymbol{B}\boldsymbol{O}^{\top}\boldsymbol{A} \tag{33}$$

where $\boldsymbol{O}$ is a random $N \times N$ orthogonal matrix drawn from the Haar measure for the orthogonal group. Our starting point is to express the above dynamical system with an integral representation of Dirac-Delta functions $1 = \int dz\delta(z) = \int \frac{dz d\chi}{2\pi i}e^{-\chi z}$ where the integration variable $\chi \in (-i\infty, i\infty)$ runs along the imaginary axis. Performing this for each time $t$ of the dynamics yields

$$Z = \int \mathcal{D}\boldsymbol{v}\mathcal{D}\boldsymbol{\chi}\left\langle \exp\left(-\int dt\boldsymbol{\chi}(t)\left[\partial_t \boldsymbol{v}(t) + \boldsymbol{M}\boldsymbol{v}(t) - \boldsymbol{v}_0\delta(t)\right]\right)\right\rangle = 1 \tag{34}$$

where $\mathcal{D}\boldsymbol{v}\mathcal{D}\boldsymbol{\chi} = \lim_{\delta t \to 0}\prod_{n=-\infty}^{\infty}\frac{d\boldsymbol{v}(n \cdot \delta t)d\boldsymbol{\chi}(n \cdot \delta t)}{2\pi i}$ is the flat measure over $\boldsymbol{v}(t), \boldsymbol{\chi}(t)$ in the continuous time limit $\delta t \to 0$. The average $\langle \cdot \rangle$ is computed over the random orthogonal matrix $\boldsymbol{O}$.

To simplify the calculation, we transform our dynamics into Fourier space

$$\boldsymbol{v}(t) \equiv \int \frac{d\omega}{2\pi}\exp\left(i\omega t\right)\boldsymbol{v}(\omega) \ , \ \boldsymbol{\chi}(t) \equiv \int \frac{d\omega}{2\pi}\exp\left(i\omega t\right)\boldsymbol{\chi}(\omega), \tag{35}$$

which transforms the original integral over $t$ into an integral over $\omega$ as $\int d\omega\boldsymbol{\chi}(\omega) \cdot [(i\omega)\boldsymbol{v}(\omega) + \boldsymbol{M}\boldsymbol{v}(\omega) - \boldsymbol{v}_0]$.

**Disorder Average** Averaging the resulting expression over the orthogonal matrix $\boldsymbol{O}$, we obtain the following representation of $Z$ as an integral over order parameters $\{\boldsymbol{\Sigma}, \boldsymbol{\Psi}\}$

$$Z = \int \mathcal{D}\boldsymbol{\Sigma}\mathcal{D}\boldsymbol{\Psi}\exp\left(-N\mathcal{S}[\boldsymbol{\Sigma}, \boldsymbol{\Psi}]\right). \tag{36}$$

In the above expression, the action $\mathcal{S}$ has the form

$$\mathcal{S}[\boldsymbol{\Sigma}, \boldsymbol{\Psi}] = -\text{Tr}\boldsymbol{\Sigma}\boldsymbol{\Psi} - \frac{1}{N}\ln\mathcal{Z}_A(\boldsymbol{\Psi}) - \text{Tr}\boldsymbol{\Sigma}\hat{\boldsymbol{\Sigma}}_{\star} - \frac{1}{N}\ln\det\left(\hat{\boldsymbol{\Sigma}}_{\star} \otimes \boldsymbol{I} + \boldsymbol{V} \otimes \boldsymbol{B}\right) + \ln\det\boldsymbol{\Sigma} \tag{37}$$

where $\hat{\boldsymbol{\Sigma}}_{\star}$ solves the implicit equation $\frac{\partial \mathcal{S}}{\partial \hat{\boldsymbol{\Sigma}}_{\star}} = -\boldsymbol{\Sigma} + \text{tr}\left(\hat{\boldsymbol{\Sigma}}_{\star} \otimes \boldsymbol{I} + \boldsymbol{V} \otimes \boldsymbol{B}\right)^{-1} = 0$ and the single site measure $\mathcal{Z}_A$ has the form

$$\mathcal{Z}_A(\boldsymbol{\Psi}) = \int \mathcal{D}\boldsymbol{v}\mathcal{D}\boldsymbol{\chi}\exp\left(-\int d\omega\boldsymbol{\chi}(\omega) \cdot \left[(i\omega)\boldsymbol{v}(z) - \boldsymbol{v}_0 - \int d\omega'\Psi_{v\chi}(\omega, \omega')\boldsymbol{A}\boldsymbol{v}(\omega')\right]\right)$$

$$\exp\left(-\frac{1}{2}\int d\omega d\omega'\left[\Psi_{vv}(\omega, \omega')\boldsymbol{v}(\omega)^{\top}\boldsymbol{A}^2\boldsymbol{v}(\omega') + \Psi_{\chi\chi}(\omega, \omega')\boldsymbol{\chi}(\omega) \cdot \boldsymbol{\chi}(\omega')\right]\right) \tag{38}$$

**Taking the Large System Size Limit**  To study the limit of $N \to \infty$, the saddle point equations over $\boldsymbol{\Sigma}$ and $\boldsymbol{\Psi}$ give

$$\frac{\partial \mathcal{S}}{\partial \boldsymbol{\Psi}(\omega, \omega')} = -\begin{bmatrix} \Sigma_{vv}(\omega, \omega') & \Sigma_{v\chi}(\omega, \omega') \\ \Sigma_{\chi v}(\omega, \omega') & \Sigma_{\chi\chi}(\omega, \omega') \end{bmatrix} + \begin{bmatrix} \frac{1}{N}\left\langle \boldsymbol{v}(\omega)^\top \boldsymbol{A}^2 \boldsymbol{v}(\omega') \right\rangle & \frac{1}{N}\left\langle \boldsymbol{v}(\omega)^\top \boldsymbol{A}\boldsymbol{\chi}(\omega') \right\rangle \\ \frac{1}{N}\left\langle \boldsymbol{\chi}(\omega)^\top \boldsymbol{A}\boldsymbol{v}(\omega') \right\rangle & \frac{1}{N}\left\langle \boldsymbol{\chi}(\omega) \cdot \boldsymbol{\chi}(\omega') \right\rangle \end{bmatrix}$$

$$\frac{\partial \mathcal{S}}{\partial \boldsymbol{\Sigma}(\omega, \omega')} = -\boldsymbol{\Psi}(\omega, \omega') - \hat{\boldsymbol{\Sigma}}_\star(\omega, \omega') + [\boldsymbol{\Sigma}^{-1}](\omega, \omega') = 0 \tag{39}$$

The average over $\langle \cdot \rangle$ represents an average over the Gaussian measure defined by $\mathcal{Z}_A$. The solution to these equations has the following block structure

$$\boldsymbol{\Sigma}(\omega, \omega') = \begin{bmatrix} \Sigma_{vv}(\omega, \omega') & \Sigma_{v\chi}(\omega, \omega') \\ \Sigma_{v\chi}(\omega, \omega') & 0 \end{bmatrix} \tag{40}$$

$$\hat{\boldsymbol{\Sigma}}(\omega, \omega') = \begin{bmatrix} 0 & \hat{\Sigma}_{v\chi}(\omega, \omega') \\ \hat{\Sigma}_{v\chi}(\omega, \omega') & \hat{\Sigma}_{\chi\chi}(\omega, \omega') \end{bmatrix} \tag{41}$$

$$\boldsymbol{\Psi}(\omega, \omega') = \begin{bmatrix} 0 & \Psi_{v\chi}(\omega, \omega') \\ \Psi_{v\chi}(\omega, \omega') & \Psi_{\chi\chi}(\omega, \omega') \end{bmatrix} \tag{42}$$

**Off Diagonal Blocks**  The off diagonal blocks decouple over different frequencies $\Sigma_{v\chi}(\omega, \omega') = \delta(\omega - \omega')\Sigma_{v\chi}(\omega)$ and satisfy the following equations

$$\Sigma_{v\chi}(\omega) = \text{tr}\boldsymbol{A}\left(i\omega + \Psi_{v\chi}(\omega)\boldsymbol{A}\right)^{-1} = \text{tr}\left(\hat{\Sigma}_{v\chi}(\omega) + \boldsymbol{B}\right)^{-1}, \tag{43}$$

$$\Psi_{v\chi}(\omega)\Sigma_{v\chi}(\omega) = 1 - \hat{\Sigma}_{v\chi}(\omega)\Sigma_{v\chi}(\omega) \tag{44}$$

Introduce the $\tau$-transform $\tau_M$ of a matrix $\boldsymbol{M}$ as

$$\tau_M(i\omega) \equiv \text{tr}\boldsymbol{M}(i\omega + \boldsymbol{M})^{-1} \tag{45}$$

as well as its inverse function $i\omega_M(\tau)$ [3]. Then our saddle point equations give us

$$\tau = \Sigma_{v\chi}(\omega)\Psi_{v\chi}(\omega) = \tau_A(i\omega_A) = \tau_B(i\omega_B) \tag{46}$$

where $i\omega_A = \frac{i\omega}{\Psi_{v\chi}(\omega)}$ and $i\omega_B = (\tau^{-1} - 1)\Psi_{v\chi}(\omega)$. Putting these equations together, we find

$$(i\omega_A(\tau))(i\omega_B(\tau)) = \frac{1 - \tau}{\tau}(i\omega) \tag{47}$$

This equation is to be solved for $\tau(\omega)$. Once this function is determined we can use it to compute the diagonal blocks of $\boldsymbol{\Sigma}, \boldsymbol{\Psi}$ as we describe in the next section.

**Diagonal Blocks**  Now, we can determine the diagonal blocks, which determine the covariance structure of the $\boldsymbol{v}(\omega)$ variables

$$\Sigma_{vv}(\omega, \omega') = \frac{1}{N}\boldsymbol{v}_0^\top \left(i\omega + \Psi_{v\chi}(\omega)\boldsymbol{A}\right)^{-1} \boldsymbol{A}^2 \left(i\omega' + \Psi_{v\chi}(\omega')\boldsymbol{A}\right)^{-1} \boldsymbol{v}_0$$

$$- \Psi_{\chi\chi}(\omega, \omega') \underbrace{\text{tr}\boldsymbol{A}^2 \left(i\omega + \Psi_{v\chi}(\omega)\boldsymbol{A}\right)^{-1} \left(i\omega' + \Psi_{v\chi}(\omega')\boldsymbol{A}\right)^{-1}}_{\mathcal{A}(\omega,\omega')}$$

$$\Sigma_{vv}(\omega, \omega') = -\hat{\Sigma}_{\chi\chi}(\omega, \omega') \underbrace{\text{tr}\left(\hat{\Sigma}_{v\chi}(\omega) + \boldsymbol{B}\right)^{-1} \left(\hat{\Sigma}_{v\chi}(\omega') + \boldsymbol{B}\right)^{-1}}_{\mathcal{B}(\omega,\omega')}$$

$$\Psi_{\chi\chi}(\omega, \omega') = -\hat{\Sigma}_{\chi\chi}(\omega, \omega') - \Sigma_{vv}(\omega, \omega')\Sigma_{v\chi}(\omega)^{-1}\Sigma_{v\chi}(\omega')^{-1} \tag{48}$$

where we introduced functions $\mathcal{A}$ and $\mathcal{B}$ which can be determined from the (already obtained) off-diagonal blocks. Combining these equations

$$\Sigma_{vv}(\omega, \omega') = \frac{\frac{1}{N}\boldsymbol{v}_0^\top \left(i\omega + \Psi_{v\chi}(\omega)\boldsymbol{A}\right)^{-1} \boldsymbol{A}^2 \left(i\omega' + \Psi_{v\chi}(\omega')\boldsymbol{A}\right)^{-1} \boldsymbol{v}_0}{1 - [\Sigma_{v\chi}(\omega)^{-1}\Sigma_{v\chi}(\omega')^{-1} - \mathcal{B}(\omega, \omega')^{-1}]\mathcal{A}(\omega, \omega')} \tag{49}$$

From this expression, both $\hat{\Sigma}_{\chi\chi}(\omega, \omega') = -\mathcal{B}(\omega, \omega')\Sigma_{vv}(\omega, \omega')$ and $\Psi_{\chi\chi}(\omega, \omega') = -\hat{\Sigma}_{\chi\chi}(\omega, \omega') - \Sigma_{vv}(\omega, \omega')\Sigma_{v\chi}(\omega)^{-1}\Sigma_{v\chi}(\omega')^{-1}$ are determined.

---

[3]Up to a change in signs and $i\omega \to -z$ this is equivalent to the $t$-transform of a random matrix (Potters & Bouchaud, 2020).

**Deterministic Equivalent**   From the functions $\Psi_{v\chi}(\omega)$ and $\Psi_{\chi\chi}(\omega, \omega')$ we obtain the following determinstic equivalent for the outer product of $v$ variables

$$\boldsymbol{v}(\omega)\boldsymbol{v}(\omega')^{\top} \simeq (i\omega + \Psi_{v\chi}(\omega)\boldsymbol{A})^{-1} \left[\boldsymbol{v}_0\boldsymbol{v}_0 - \Psi_{\chi\chi}(\omega, \omega')\boldsymbol{I}\right] (i\omega' + \Psi_{v\chi}(\omega')\boldsymbol{A})^{-1} \quad (50)$$

where the $\simeq$ expression holds under trace against a test matrix (i.e. $\boldsymbol{M}_1 \simeq \boldsymbol{M}_2 \implies \operatorname{tr}\boldsymbol{C}\boldsymbol{M}_1 = \operatorname{tr}\boldsymbol{C}\boldsymbol{M}_2$) (Potters & Bouchaud, 2020). This function will be used in subsequent expressions to give an exact result for the loss landscape of our randomly rotated ICL loss function. Approximations of this will give rise to our scaling law results as we discuss in Appendix F.5.

## B   MODEL DEFINITION, MASKING MECHANICS, REDUCED-$\Gamma$ MODEL

### B.1   LINEAR ATTENTION AND POSITIONAL MASKING FOR OUR TASK

In this section, we provide more detail on the structure of the masking used in our task. First, we note that the target values $y_\mu$ are only provided for the $P$ labeled examples within each context matrix. Second, we note that the residual attention operations are masked differently for the $P$ labeled examples and the $K$ evaluation points. To define our masking operation precisely, we introduce the notation $\mathbf{1}_k \in \mathbb{R}^k$ as a vector consisting of all $k$ entries equal to one. We introduce the masking matrix $\boldsymbol{M} \in \mathbb{R}^{(P+K)\times(P+K)}$ for the residual stream which has the following block structure

$$\boldsymbol{M} = \begin{bmatrix} -\mathbf{1}_P \, \mathbf{1}_P^{\top} & \mathbf{0} \\ \mathbf{1}_K \, \mathbf{1}_P^{\top} & \mathbf{0} \end{bmatrix}. \quad (51)$$

Now that this positional masking matrix is introduced, we can conveniently express our update rule for the residual stream

$$\boldsymbol{h}_\mu^{\ell+1} = \boldsymbol{h}_\mu^{\ell} + \frac{1}{PL} \sum_{\nu=1}^{P+K} M_{\mu\nu} \left(\boldsymbol{k}_\nu^{\ell} \cdot \boldsymbol{q}_\mu^{\ell}\right) \boldsymbol{v}_\nu^{\ell}$$

$$= \boldsymbol{h}_\mu^{\ell} + \frac{1}{PL} \sum_{\nu=1}^{P} M_{\mu\nu} \left(\boldsymbol{k}_\nu^{\ell} \cdot \boldsymbol{q}_\mu^{\ell}\right) \boldsymbol{v}_\nu^{\ell} \, , \, \mu \in [P+K] \quad (52)$$

where $\boldsymbol{k}_\mu^{\ell} = \boldsymbol{W}_k \cdot \boldsymbol{h}_\mu^{\ell}, \boldsymbol{q}_\mu^{\ell} = \boldsymbol{W}_q \boldsymbol{h}_\mu^{\ell}, \boldsymbol{v}_\mu^{\ell} = \boldsymbol{W}_v^{\ell} \boldsymbol{h}_\mu^{\ell}$ and we dropped the sum over evaluation points $\{P + 1, ..., P + K\}$ due to the structure of the positional mask $\boldsymbol{M}$. We thus see that the masked update rule has two properties

1. It prevents the test points $\boldsymbol{x}_\mu$ for $\mu \in \{P + 1, ..., P + K\}$ from being used in the residual stream updates. Only the first $P$ training points are utilized.

2. It provides an opposite sign for the updates on training points and on testing points. We will see that this will enable the model to implement an in-context gradient descent rule. In such a rule, a subspace of the residual variables will encode residual errors $y_\mu - f_\mu^{\ell}$ for the training predictions and $+f_\mu^{\ell}$ for the test points $\mu \in \{P + 1, .., P + K\}$. Instead, one could use the same signs for training points and test points in $\boldsymbol{M}$ and simply negate the output of the model at the end $f \to -f$.

### B.2   DERIVATION OF REDUCED GAMMA MODEL

In this section, we describe the conditions under which a linear attention transformer model can be reparameterized as the recurrent reduced-$\Gamma$ model we discuss in the main text.

**Alignment Assumptions**   Following Von Oswald et al. (2023); Zhang et al. (2025); Lu et al. (2025), we study configurations of weights that encode information about inputs $\boldsymbol{x}$ and targets $y$ in orthogonal subspaces of the residual stream. Concretely, the following assumptions are made on the input weights, which implement in-context preconditioned gradient descent

$$\boldsymbol{W}_x^{\top} \boldsymbol{w}_y = 0 \, , \, \boldsymbol{W}_x^{\top} \boldsymbol{w}_o = 0 \, , \, \boldsymbol{w}_y \cdot \boldsymbol{w}_y = |\boldsymbol{w}_y||\boldsymbol{w}_y|. \quad (53)$$

Collectively these assumptions imply that read-in weights for the targets $\boldsymbol{w}_y$ and readout weights $\boldsymbol{w}_o$ perfectly align and that the read-in weights for $\boldsymbol{x}$ $\boldsymbol{W}_x$ project input data to an orthogonal subspace. Next we study the following set of alignment conditions for the key, query and value matrices

$$\boldsymbol{W}_x^\top \left(\boldsymbol{W}_k^\ell\right)^\top \boldsymbol{W}_q^\ell \boldsymbol{W}_x \propto \boldsymbol{\Gamma}^\ell \in \mathbb{R}^{D \times D} \tag{54}$$

$$\boldsymbol{W}_v \boldsymbol{W}_x = 0 \ , \ \boldsymbol{W}_v \boldsymbol{w}_y \propto \boldsymbol{w}_y \tag{55}$$

Under these assumptions, we can define a collection of $D \times D$ matrices $\boldsymbol{\Gamma}^\ell$

$$\boldsymbol{\Gamma}^\ell \equiv \left(\boldsymbol{w}_o^\top \boldsymbol{W}_v^\ell \boldsymbol{w}_y\right) \boldsymbol{W}_x^\top (\boldsymbol{W}_k^\ell)^\top (\boldsymbol{W}_q^\ell) \boldsymbol{W}_x \in \mathbb{R}^{D \times D}, \tag{56}$$

which gives rise to the following residual stream dynamics for $\Delta_\mu^\ell \equiv \boldsymbol{w}_o \cdot \boldsymbol{h}^\ell$

$$\Delta_\mu^{\ell+1} = \Delta_\mu^\ell + \frac{1}{LP} \sum_{\nu=1}^P M_{\mu\nu} \boldsymbol{x}_\nu^\top \boldsymbol{\Gamma}^\ell \boldsymbol{x}_\mu \Delta_\nu^\ell. \tag{57}$$

We note that this can be separated into two distinct dynamical systems, one for the first $P$ training points

$$\Delta_\mu^{\ell+1} = \Delta_\mu^\ell - \frac{1}{LP} \sum_{\nu=1}^P \boldsymbol{x}_\nu^\top \boldsymbol{\Gamma}^\ell \boldsymbol{x}_\mu \Delta_\nu^\ell \ , \ \mu \in \{1, ..., P\} \tag{58}$$

which form a closed dynamical system on the $P$ labeled training points. From these $P$ variables $\{\Delta_\mu^\ell\}_{\mu \in [P]}$, we can describe how the remaining $K$ points evolve

$$\Delta_\mu^{\ell+1} = \Delta_\mu^\ell + \frac{1}{LP} \sum_{\nu=1}^P \boldsymbol{x}_\nu^\top \boldsymbol{\Gamma}^\ell \boldsymbol{x}_\mu \Delta_\nu^\ell = \frac{1}{LP} \sum_{\ell=1}^L \sum_{\nu=1}^P \boldsymbol{x}_\nu^\top \boldsymbol{\Gamma}^\ell \boldsymbol{x}_\mu \Delta_\nu^\ell \ , \ \mu \in \{P+1, ..., P+K\}. \tag{59}$$

**Recurrence** Instead of defining separate $\boldsymbol{\Gamma}^\ell$ matrices for each layer $\ell$, we instead can examine a recurrent model where the weights are tied $\boldsymbol{\Gamma}^\ell = \boldsymbol{\Gamma}$ for all $\ell \in [L]$. We relax this assumption in Appendix G.1 and in many of the settings we analyze recurrence has no impact compared to decoupling layers. Under this constraint, the residual stream of the model has the following dynamics

$$\Delta_\mu^{\ell+1} = \Delta_\mu^\ell - \frac{1}{LP} \sum_{\nu=1}^P \boldsymbol{x}_\mu^\top \boldsymbol{\Gamma} \boldsymbol{x}_\nu \ \Delta_\nu^\ell \ , \ \mu \in \{1, ..., P\}$$

$$\Delta_\mu^{\ell+1} = \Delta_\mu^\ell + \frac{1}{LP} \sum_{\nu=1}^P \boldsymbol{x}_\mu^\top \boldsymbol{\Gamma} \boldsymbol{x}_\nu \ \Delta_\nu^\ell \ , \ \mu \in \{P+1, ..., P+K\} \tag{60}$$

Let $\boldsymbol{\Delta}^\ell \in \mathbb{R}^P$ represent the residual stream variables restricted to the $P$ unmasked training points in the context. This vector has the residual stream dynamics

$$\boldsymbol{\Delta}^\ell = \left(\boldsymbol{I} - \frac{1}{LP} \boldsymbol{X}^\top \boldsymbol{\Gamma} \boldsymbol{X}\right)^\ell \boldsymbol{y}. \tag{61}$$

However, the recursion is different for the test set since these points only receive attention signals from the $P$ unmasked training tokens. For one of the test points $\boldsymbol{x}_\star$, the prediction of the model $f_\star$ satisfies

$$f_\star = \frac{1}{LP} \boldsymbol{x}_\star^\top \boldsymbol{\Gamma} \boldsymbol{X} \sum_{\ell=0}^{L-1} \boldsymbol{h}^\ell = \frac{1}{LP} \boldsymbol{x}_\star^\top \boldsymbol{\Gamma} \boldsymbol{X} \sum_{\ell=0}^{L-1} \left(\boldsymbol{I} - \frac{1}{LP} \boldsymbol{X}^\top \boldsymbol{\Gamma} \boldsymbol{X}\right)^\ell \boldsymbol{y} \tag{62}$$

**Equivalence to Preconditioned In-Context Gradient Descent** This update rule is equivalent to implementing preconditioned in-context gradient descent steps for a linear regression model $f = \frac{1}{\sqrt{D}} \boldsymbol{\beta} \cdot \boldsymbol{x}$. To see this, define an in-context training loss $\hat{\mathcal{L}}(\boldsymbol{\beta}, \boldsymbol{D})$ on the $P$ labeled training points for context $\boldsymbol{D}$

$$\hat{\mathcal{L}}(\boldsymbol{D}) = \frac{1}{2P} ||D^{-1/2} \boldsymbol{X}^\top \boldsymbol{\beta} - \boldsymbol{y}||^2 \tag{63}$$

Preconditioned gradient descent with learning rate $D/L$ and a preconditioner matrix $\mathbf{\Gamma}$ generates the following dynamics on the learned ICL weights $\boldsymbol{\beta}$

$$\boldsymbol{\beta}^{\ell+1} = \boldsymbol{\beta}^\ell - L^{-1}\mathbf{\Gamma}\,\nabla\mathcal{L}(\boldsymbol{\beta}^\ell, \boldsymbol{D}) = \boldsymbol{\beta}^\ell - \frac{\sqrt{D}}{LP}\mathbf{\Gamma}\boldsymbol{X}\left(D^{-1/2}\boldsymbol{X}^\top\boldsymbol{\beta}^\ell - \boldsymbol{y}\right) \tag{64}$$

Defining a variable $\boldsymbol{h}^\ell \equiv \boldsymbol{y} - D^{-1/2}\boldsymbol{X}^\top\boldsymbol{\beta}^\ell$, we note that this variable satisfies the same recursion as the residual stream variables $\boldsymbol{h}^\ell$ on the training points described above which gives the identical solution $\boldsymbol{h}^\ell = \left(\boldsymbol{I} - \frac{1}{LP}\boldsymbol{X}^\top\mathbf{\Gamma}\boldsymbol{X}\right)^\ell\boldsymbol{y}$. The function $f_\star$ on a test point $\boldsymbol{x}_\star$ takes the form

$$f_\star = \frac{1}{\sqrt{D}}\boldsymbol{\beta}^L \cdot \boldsymbol{x}_\star = \frac{1}{LP}\boldsymbol{x}_\star^\top\mathbf{\Gamma}\boldsymbol{X}\sum_{\ell=0}^{L-1}\boldsymbol{h}^\ell = \frac{1}{LP}\boldsymbol{x}_\star^\top\mathbf{\Gamma}\boldsymbol{X}\sum_{\ell=0}^{L-1}\left(\boldsymbol{I} - \frac{1}{LP}\boldsymbol{X}^\top\mathbf{\Gamma}\boldsymbol{X}\right)^\ell\boldsymbol{y}. \tag{65}$$

**Test Error Formula for Linear Models**   We now desire to compute the expected test loss (averaged over the dataset $\boldsymbol{X}$ and noise $\boldsymbol{\epsilon}$) for (noisy) linear target function $y(\boldsymbol{x})$

$$y(\boldsymbol{x}) = \frac{1}{\sqrt{D}}\boldsymbol{\beta}_\star \cdot \boldsymbol{x} + \sigma\epsilon\,,\ \left\langle\boldsymbol{x}\boldsymbol{x}^\top\right\rangle = \mathbf{\Sigma}\,,\ \left\langle\epsilon^2\right\rangle = 1 \tag{66}$$

where $\mathbf{\Sigma}$ is the (population) covariance of the inputs. The expected loss under this assumption is

$$\mathcal{L} = \frac{1}{D}\left\langle\left(\boldsymbol{\beta}^L - \boldsymbol{\beta}_\star\right)^\top\mathbf{\Sigma}\left(\boldsymbol{\beta}^L - \boldsymbol{\beta}_\star\right)\right\rangle_{\boldsymbol{X},\boldsymbol{\epsilon}} + \sigma^2 \tag{67}$$

Letting $\boldsymbol{\epsilon} \in \mathbb{R}^P$ represent the noise on the $P$ labeled training points, the target weights have the form

$$\begin{aligned}
\boldsymbol{\beta}^L &= \frac{\sqrt{D}}{LP}\mathbf{\Gamma}\boldsymbol{X}\sum_{\ell=0}^{L}\left(\boldsymbol{I} - \frac{1}{LP}\boldsymbol{X}^\top\mathbf{\Gamma}\boldsymbol{X}\right)^\ell\left[\frac{1}{\sqrt{D}}\boldsymbol{X}^\top\boldsymbol{\beta}_\star + \sigma\boldsymbol{\epsilon}\right]\\
&= L^{-1}\mathbf{\Gamma}\hat{\mathbf{\Sigma}}\sum_{\ell=0}^{L-1}\left(\boldsymbol{I} - L^{-1}\mathbf{\Gamma}\hat{\mathbf{\Sigma}}\right)^\ell\boldsymbol{\beta}_\star + \frac{\sigma\sqrt{D}}{LP}\,\mathbf{\Gamma}\sum_{\ell=0}^{L-1}\left(\boldsymbol{I} - L^{-1}\hat{\mathbf{\Sigma}}\mathbf{\Gamma}\right)^\ell\boldsymbol{X}\boldsymbol{\epsilon}\\
&= \boldsymbol{\beta}_\star - \left(\boldsymbol{I} - L^{-1}\mathbf{\Gamma}\hat{\mathbf{\Sigma}}\right)^L\boldsymbol{\beta}_\star + \frac{\sigma\sqrt{D}}{LP}\mathbf{\Gamma}\sum_{\ell=0}^{L-1}\left(\boldsymbol{I} - L^{-1}\hat{\mathbf{\Sigma}}\mathbf{\Gamma}\right)^\ell\boldsymbol{X}\boldsymbol{\epsilon}
\end{aligned} \tag{68}$$

where we defined $\hat{\mathbf{\Sigma}} = \frac{1}{P}\boldsymbol{X}\boldsymbol{X}^\top \in \mathbb{R}^{D\times D}$. Then we note that for $\alpha \equiv P/D$ that the reducible loss $\mathcal{L} - \sigma^2$ has the form

$$\begin{aligned}
\mathcal{L} - \sigma^2 &= \frac{1}{D}\left\langle\left(\boldsymbol{\beta}_\star - \boldsymbol{\beta}^L\right)^\top\mathbf{\Sigma}\left(\boldsymbol{\beta}_\star - \boldsymbol{\beta}^L\right)\right\rangle\\
&= \frac{1}{D}\boldsymbol{\beta}_\star^\top\left\langle\left(\boldsymbol{I} - L^{-1}\mathbf{\Gamma}\hat{\mathbf{\Sigma}}\right)^{L\top}\mathbf{\Sigma}\left(\boldsymbol{I} - L^{-1}\mathbf{\Gamma}\hat{\mathbf{\Sigma}}\right)^L\right\rangle\boldsymbol{\beta}_\star\\
&\quad + \frac{\sigma^2}{\alpha L^2}\operatorname{Tr}\mathbf{\Gamma}^\top\mathbf{\Gamma}\sum_{\ell,\ell'=0}^{L-1}\left\langle\left(\boldsymbol{I} - L^{-1}\mathbf{\Gamma}\hat{\mathbf{\Sigma}}\right)^\ell\hat{\mathbf{\Sigma}}\left[\left(\boldsymbol{I} - L^{-1}\mathbf{\Gamma}\hat{\mathbf{\Sigma}}\right)^{\ell'}\right]^\top\right\rangle
\end{aligned} \tag{69}$$

In the next sections, we will compute this quantity for various distributions for $\hat{\mathbf{\Sigma}}$ and $\boldsymbol{\beta}_\star$.

## C   ISOTROPIC (**ISO**) SETTING THEORY

In this section, we consider isotropic covariates and tasks so that the data covariance $\mathbf{\Sigma}$ and the task vector covariance $\mathbf{\Omega}$ are both identity

$$\mathbf{\Sigma} = \left\langle\boldsymbol{x}\boldsymbol{x}^\top\right\rangle = \boldsymbol{I}\,,\ \mathbf{\Omega} = \left\langle\boldsymbol{\beta}_\star\boldsymbol{\beta}_\star^\top\right\rangle = \boldsymbol{I}. \tag{70}$$

where the average for target vectors $\boldsymbol{\beta}_\star$ is over different contexts. We further operate in the *high dimensional* asymptotic limit where both context length $P$ and input dimension $D$ diverge with fixed ratio

$$P, D \to \infty\,,\ P/D = \alpha. \tag{71}$$

In this setting, the spectrum of the empirical covariance matrix $\hat{\mathbf{\Sigma}} = \frac{1}{P}\boldsymbol{X}\boldsymbol{X}^\top$ follows the well-known Marchenko-Pastur law, where the eigenvalue density $\rho(\lambda)$ depends explicitly on the aspect ratio $\alpha$ (Potters & Bouchaud, 2020; Advani et al., 2020). We will first describe an asymptotic limit of SGD for the shallow $L = 1$ case where the dynamics on the matrix $\mathbf{\Gamma}$ are linear. We will then pursue a description of gradient flow dynamics in an arbitrary depth $L$ model.

## C.1 SHALLOW SGD THEORY

In this section, we consider the SGD dynamics of the shallow architecture $L = 1$. This excercise will establish at what rate our dynamics will converge to gradient flow and how finite batchsize $B$, context length $P$ and number of masked evaluation points $K$ impact the SGD dynamics. In this case, the updates have the form

$$\boldsymbol{\Gamma}(t+1) = \boldsymbol{\Gamma}(t) + \eta\,\boldsymbol{G}(t)$$

$$\boldsymbol{G}(t) = \frac{1}{B}\sum_n \hat{\boldsymbol{\Sigma}}_{\star,n}\left[(\boldsymbol{I} - \boldsymbol{\Gamma}\boldsymbol{\Sigma}_n)\boldsymbol{\beta}_n - \sigma P^{-1}\sqrt{D}\boldsymbol{\Gamma}\boldsymbol{X}_n\boldsymbol{\epsilon}_n\right]\left[\boldsymbol{\Sigma}_n\boldsymbol{\beta}_n + \sigma P^{-1}\sqrt{D}\boldsymbol{X}_n\boldsymbol{\epsilon}_n\right]^\top$$

$$\hat{\boldsymbol{\Sigma}}_n = \frac{1}{P}\boldsymbol{X}_n^\top\boldsymbol{X}_n\,,\ \hat{\boldsymbol{\Sigma}}_{\star,n} = \frac{1}{K}\boldsymbol{X}_{\star,n}^\top\boldsymbol{X}_{\star,n} \tag{72}$$

where $\boldsymbol{X} \in \mathbb{R}^{P\times D}$ is the set of training points and $\boldsymbol{X}_\star \in \mathbb{R}^{K\times D}$ represents the set of $K$ evaluation points and $\boldsymbol{\epsilon} \in \mathbb{R}^P$ represents the label noise on the provided targets $\boldsymbol{y}$. We remind the reader that we will operate in the joint scaling limit

$$P, B, K, D \to \infty\,,\ P/D = \alpha\,,\ K/D = \kappa\,,\ B/D = \tau \tag{73}$$

The population ICL loss has the form

$$\mathcal{L}(t) = \frac{1}{D}\left\langle|\boldsymbol{\Gamma}(t) - \boldsymbol{I}|^2\right\rangle, \tag{74}$$

where the average is performed over all possible draws of data and tasks. This function admits the recursion

$$\mathcal{L}(t+1) = \mathcal{L}(t) - 2\eta\mathrm{tr}[\boldsymbol{\Gamma}(t) - \boldsymbol{I}]^\top\langle\boldsymbol{G}(t)\rangle + \eta^2\mathrm{tr}\left\langle\boldsymbol{G}(t)^\top\boldsymbol{G}(t)\right\rangle \tag{75}$$

We have the following mean for the gradient

$$\langle\boldsymbol{G}(t)\rangle = (\boldsymbol{I} - (1 + \alpha^{-1} + \sigma^2\alpha^{-1})\boldsymbol{\Gamma}) \tag{76}$$

Thus if we only updated using mean gradients the model would relax to a fixed point $\boldsymbol{\Gamma}_\star = \left(1 + \alpha^{-1} + \sigma^2\alpha^{-1}\right)^{-1}\boldsymbol{I}$. However, we will see momentarily that SGD noise in the gradients impact the $\boldsymbol{\Gamma}$ that SGD converges to

$$\mathrm{tr}\left\langle\boldsymbol{G}(t)^\top\boldsymbol{G}(t)\right\rangle = \frac{1}{B^2}\sum_{nm}\mathrm{tr}\left\langle\boldsymbol{u}_n\boldsymbol{v}_n^\top\boldsymbol{v}_m\boldsymbol{u}_m^\top\right\rangle \tag{77}$$

where $\boldsymbol{u}_n = \hat{\boldsymbol{\Sigma}}_{\star,n}\left[(\boldsymbol{I} - \boldsymbol{\Gamma}\boldsymbol{\Sigma}_n)\boldsymbol{\beta}_n - \sigma p^{-1}\sqrt{d}\boldsymbol{\Gamma}\boldsymbol{X}_n\boldsymbol{\epsilon}_n\right]$ and $\boldsymbol{v}_n = \boldsymbol{\Sigma}_n\boldsymbol{\beta}_n + \sigma p^{-1}\sqrt{d}\boldsymbol{X}_n\boldsymbol{\epsilon}_n$. Doing the averages

$$\frac{1}{D}\langle\boldsymbol{u}_n\cdot\boldsymbol{u}_m\rangle = \delta_{mn}\mathrm{tr}\left[\boldsymbol{I} - 2\boldsymbol{\Gamma} + (1 + \alpha^{-1})\boldsymbol{\Gamma}^2 + \sigma^2\alpha^{-1}\boldsymbol{\Gamma}^2\right](1 + \kappa^{-1}) \tag{78}$$

$$= \delta_{mn}(1 + \alpha^{-1}(1 + \sigma^2))(1 + \kappa^{-1})\mathrm{tr}\left[(\boldsymbol{\Gamma} - \boldsymbol{\Gamma}_\star)^2 + \frac{\alpha^{-1}(1 + \sigma^2)}{(1 + \alpha^{-1}(1 + \sigma^2))^2}\boldsymbol{I}\right] \tag{79}$$

$$\frac{1}{D}\langle\boldsymbol{v}_n\cdot\boldsymbol{v}_m\rangle = \delta_{mn}(1 + \alpha^{-1}(1 + \sigma^2)) \tag{80}$$

From the above results, we get the following second moment structure

$$\mathrm{tr}\left\langle\boldsymbol{G}(t)^\top\boldsymbol{G}(t)\right\rangle = \mathrm{tr}\left\langle\boldsymbol{G}(t)\right\rangle^\top\langle\boldsymbol{G}(t)\rangle$$
$$+ \frac{1}{\tau}(1 + \kappa^{-1})(1 + \alpha^{-1}(1 + \sigma^2))^2\left[(\boldsymbol{\Gamma} - \boldsymbol{\Gamma}_\star)^2 + \frac{\alpha^{-1}(1 + \sigma^2)}{(1 + \alpha^{-1}(1 + \sigma^2))^2}\boldsymbol{I}\right]. \tag{81}$$

Thus we get a recursion

$$\boldsymbol{\Gamma}(t+1) = \boldsymbol{\Gamma}(t) + \eta(1 + \alpha^{-1}(1 + \sigma^2))\left[\boldsymbol{\Gamma}_\star - \boldsymbol{\Gamma}(t)\right] + \eta\,\boldsymbol{\Xi}(t) \tag{82}$$

where $\Xi \in \mathbb{R}^{D \times D}$ is the noise process with variance given above. We define a quantity $C(t)$ which measures the relaxation of $\Gamma$ to $\Gamma_\star$

$$C(t) = \frac{1}{D}|\Gamma(t) - \Gamma_\star|^2 \ , \ \Gamma_\star = \left[1 + \alpha^{-1}(1 + \sigma^2)\right]^{-1} I \tag{83}$$

This function exhibits the following linear dynamics

$$C(t+1) = \left[1 - \eta\left(1 + \alpha^{-1}(1 + \sigma^2)\right)\right]^2 C(t)$$
$$+ \frac{\eta^2}{\tau}(1 + \kappa^{-1})(1 + \alpha^{-1}(1 + \sigma^2))^2 \left[C(t) + \frac{\alpha^{-1}(1 + \sigma^2)}{(1 + \alpha^{-1}(1 + \sigma^2))^2}\right] \tag{84}$$

This recursion again takes the form

$$C(t+1) = a(\eta, \alpha, \kappa, \tau)C(t) + b(\eta, \alpha, \kappa, \tau)$$

$$a(\eta, \alpha, \kappa, \tau) = \left[1 - \eta\left(1 + \alpha^{-1}(1 + \sigma^2)\right)\right]^2 + \frac{\eta^2}{\tau}(1 + \kappa^{-1})(1 + \alpha^{-1}(1 + \sigma^2))^2$$

$$b(\eta, \alpha, \kappa, \tau) = \frac{\eta^2}{\tau}(1 + \kappa^{-1})\alpha^{-1}(1 + \sigma^2) \tag{85}$$

Now we need to compute the ICL loss

$$\mathcal{L}(t) = \mathrm{tr}[\Gamma(t) - \Gamma_\star + \Gamma_\star - I]^2 = C(t) + 2\mathrm{tr}(\Gamma_\star - \Gamma(t))(I - \Gamma_\star) + \mathrm{tr}(\Gamma_\star - I)^2$$
$$= C(t) + \frac{2\alpha^{-1}(1 + \sigma^2)}{(1 + \alpha^{-1}(1 + \sigma^2))^2}\left[1 - \eta(1 + \alpha^{-1}(1 + \sigma^2))\right]^t + \frac{(1 + \sigma^2)^2}{(\alpha + 1 + \sigma^2)^2} \tag{86}$$

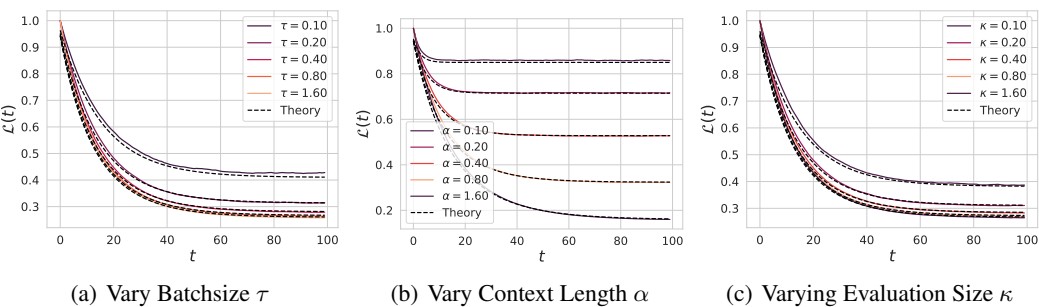

(a) Vary Batchsize $\tau$    (b) Vary Context Length $\alpha$    (c) Varying Evaluation Size $\kappa$

Figure 7: Pretraining SGD loss dynamics in the shallow $L = 1$ reduced $\Gamma$ model for (a) $\alpha, \kappa = 1$ and varying $\tau$ (b) varying $\alpha$ with $\tau = \kappa = 1$ (c) varying $\kappa$ with $\alpha, \tau = 1$. The loss monotonically improves with all three quantities $\tau, \alpha, \kappa$ and is well predicted by the asymptotic theory.

Putting this all together, we summarize our findings below.

---

**Result 10: Online SGD Dynamics for Shallow ICL with Isotropic Covariates**

Consider the reduced $\Gamma$ model with a single layer $L = 1$ and proportional asymptotics

$$P, K, B, D \to \infty \ , \ P/D = \alpha \ , \ K/D = \kappa \ , \ B/D = \tau \tag{87}$$

Then the ICL test loss $\mathcal{L}(t) = \frac{1}{d}||\Gamma(t) - I||^2$ after $t$ SGD iterations is

$$\mathcal{L}(t) = C(t) + \frac{2\alpha^{-1}(1 + \sigma^2)}{(1 + \alpha^{-1}(1 + \sigma^2))^2}\left[1 - \eta(1 + \alpha^{-1}(1 + \sigma^2))\right]^t + \frac{(1 + \sigma^2)^2}{(\alpha + 1 + \sigma^2)^2} \tag{88}$$

$$C(t) = a(\eta, \alpha, \tau, \kappa)^t + \frac{1 - a(\eta, \alpha, \tau, \kappa)^t}{1 - a(\eta, \alpha, \tau, \kappa)}b(\eta, \alpha, \tau, \kappa) \tag{89}$$

where $a(\eta, \alpha, \kappa, \tau) = \left[1 - \eta\left(1 + \alpha^{-1}(1 + \sigma^2)\right)\right]^2 + \frac{\eta^2}{\tau}(1 + \kappa^{-1})(1 + \alpha^{-1}(1 + \sigma^2))^2$ and $b(\eta, \alpha, \kappa, \tau) = \frac{\eta^2}{\tau}(1 + \kappa^{-1})\alpha^{-1}(1 + \sigma^2)$ capture the dependence on batchsize through $\tau$.

---

We verify the validity of these learning curves for shallow pretraining dynamics in Figure 7.

We thus see that deviations from gradient flow in the $L = 1$ model come in at a scale of $\eta/\tau$. The purpose of this result is to stress that only $\mathcal{O}(D^2)$ total tokens are required (unlike the prior work of Lu et al. (2025) which required $\mathcal{O}(D^3)$ tokens) for an ICL learner in this regime [4]. We verify these theoretical learning curves in Figure 7.

We also indicate the following facts about SGD effects

- The number of masked evaluation points per batch $\kappa \equiv K/D$ only alters the SGD noise terms (the terms involving $1/\tau$) and its effect vanishes in the gradient flow limit $\eta/\tau \to 0$.

- Unlike offline training, in this online SGD setting the model regularizes the final value of $\gamma(t) = $ based on label noise $\sigma^2$ to reduce overfitting.

- SGD fluctuations can impact the final loss through $\lim_{t\to\infty} C(t) = \frac{b(\eta,\alpha,\tau,\kappa)}{1-a(\eta,\alpha,\tau,\kappa)}$.

One nice consequence of this result is that the gradient flow limit can be accessed simply by controlling the scale of $\eta/\tau$ in a limit where batch size is *linear* in the dimension $B = \tau D$.

## C.2 GRADIENT FLOW IN DEEP MODELS

From the previous section, we saw that in the absence of SGD noise, gradient dynamics generated an isotropic $\boldsymbol{\Gamma}$ matrix. This fact is true for larger depth $L \geq 1$ as well. The gradient flow dynamics generate is $\boldsymbol{\Gamma}(t) = \gamma(t)\boldsymbol{I}$. To see this, we note that the gradient has the form

$$
\begin{aligned}
\frac{\partial}{\partial \boldsymbol{\Gamma}}\mathcal{L} = &-\left\langle \frac{2}{L}\sum_{\ell=1}^{L}\left[\left(\boldsymbol{I}-L^{-1}\boldsymbol{\Gamma}\hat{\boldsymbol{\Sigma}}\right)^{\ell}\right]^{\top}\hat{\boldsymbol{\Sigma}}\left[\left(\boldsymbol{I}-L^{-1}\boldsymbol{\Gamma}\hat{\boldsymbol{\Sigma}}\right)^{L-1-\ell}\right]^{\top}\left(\boldsymbol{I}-L^{-1}\boldsymbol{\Gamma}\hat{\boldsymbol{\Sigma}}\right)^{L}\right\rangle \\
&+\frac{2\sigma^2}{\alpha L^2}\boldsymbol{\Gamma}\sum_{\ell,\ell'}\left\langle \left(\boldsymbol{I}-L^{-1}\boldsymbol{\Gamma}\hat{\boldsymbol{\Sigma}}\right)^{\ell}\hat{\boldsymbol{\Sigma}}\left[\left(\boldsymbol{I}-L^{-1}\boldsymbol{\Gamma}\hat{\boldsymbol{\Sigma}}\right)^{\ell'}\right]^{\top}\right\rangle \\
&+\frac{2\sigma^2}{\alpha L^3}\boldsymbol{\Gamma}^{\top}\boldsymbol{\Gamma}\sum_{\ell,\ell',k}\left\langle \left(\boldsymbol{I}-L^{-1}\boldsymbol{\Gamma}\hat{\boldsymbol{\Sigma}}\right)^{k}\hat{\boldsymbol{\Sigma}}\left(\boldsymbol{I}-\boldsymbol{\Gamma}\hat{\boldsymbol{\Sigma}}\right)^{\ell-1-k}\hat{\boldsymbol{\Sigma}}\left[\left(\boldsymbol{I}-L^{-1}\boldsymbol{\Gamma}\hat{\boldsymbol{\Sigma}}\right)^{\ell'}\right]^{\top}\right\rangle \quad (90)
\end{aligned}
$$

Now, evaluating this at $\boldsymbol{\Gamma} = \gamma\boldsymbol{I}$ we find

$$
\begin{aligned}
\frac{\partial}{\partial \boldsymbol{\Gamma}}\mathcal{L}|_{\boldsymbol{\Gamma}=\gamma\boldsymbol{I}} = &-2\left\langle \hat{\boldsymbol{\Sigma}}\left(\boldsymbol{I}-L^{-1}\gamma\hat{\boldsymbol{\Sigma}}\right)^{2L-1}\right\rangle + \frac{2\sigma^2\gamma}{\alpha L^2}\left\langle \hat{\boldsymbol{\Sigma}}\sum_{\ell\ell'}(1-L^{-1}\gamma\hat{\boldsymbol{\Sigma}})^{\ell+\ell'}\right\rangle \\
&+\frac{2\sigma^2\gamma^2}{\alpha L^2}\sum_{\ell,\ell'}\left\langle \hat{\boldsymbol{\Sigma}}\left(\boldsymbol{I}-L^{-1}\gamma\hat{\boldsymbol{\Sigma}}\right)^{\ell+\ell'-1}\right\rangle \propto \boldsymbol{I} \quad (91)
\end{aligned}
$$

where we recognize through rotational invariance, that the above averages are proportional to the identity. Thus if $\boldsymbol{\Gamma}$ is currently in an isotropic configuration, it will remain in one throughout gradient flow. Further, we can compute the scalar $\gamma(t) = \text{tr}\boldsymbol{\Gamma}(t)$ under gradient flow $\partial_t\boldsymbol{\Gamma}(t) = -\frac{1}{2}\partial_{\boldsymbol{\Gamma}}\mathcal{L}(\boldsymbol{\Gamma})$.

$$
\frac{d}{dt}\gamma(t) = -\frac{\partial}{\partial \gamma}\text{tr}\left[\left(\boldsymbol{I}-L^{-1}\gamma\hat{\boldsymbol{\Sigma}}\right)^{2L} + \frac{\sigma^2}{\alpha}\hat{\boldsymbol{\Sigma}}^{-1}\left[\boldsymbol{I}-\left(\boldsymbol{I}-L^{-1}\gamma\hat{\boldsymbol{\Sigma}}\right)^{L}\right]^2\right] \quad (92)
$$

Now, we note that this expression can be rewritten in terms of the bulk Marchenko-Pastur eigenvalue density $\rho(\lambda) = \frac{1}{D}\sum_{k=1}^{D}\langle\delta(\lambda-\lambda_k)\rangle = \frac{\alpha}{2\pi\lambda}\sqrt{(\lambda_+ - \lambda)(\lambda - \lambda_-)}$ where $\lambda_\pm = (1\pm\alpha^{-1/2})^2$ as

$$
\frac{d}{dt}\gamma(t) = -\frac{\partial}{\partial \gamma}\int d\lambda\rho(\lambda)\left[\left(1-L^{-1}\gamma\lambda\right)^{2L} + \frac{\sigma^2}{\alpha\lambda}\left[1-\left(1-L^{-1}\gamma\lambda\right)^{L}\right]^2\right] \quad (93)
$$

which shows that we can view the dynamics as a gradient flow on a one dimensional loss landscape.

---

[4]The gap is caused by suboptimal scaling of the number of evaluation points $K$ per context (usually $K = 1$ in prior works). Note realistic LLMs get multiple error signals per context (one per token).

## D  FIXED AND STRUCTURED COVARIANCE (**FS**) SETTING

In this section we discuss the case where the data covariance is structured but potentially structured. We consider the noise free setting and $P/D \to \infty$ for simplicity where the loss has the form

$$\mathcal{L} = \text{tr } \mathbf{\Omega} \left\langle \left[ \left( \mathbf{I} - L^{-1}\mathbf{\Gamma}\mathbf{\Sigma} \right)^L \right]^\top \mathbf{\Sigma} \left( \mathbf{I} - L^{-1}\mathbf{\Gamma}\mathbf{\Sigma} \right)^L \right\rangle \tag{94}$$

In this case, gradient flow will generate dynamics that cause $\mathbf{\Gamma}$ to pick up anisotropy from the structure of $\mathbf{\Omega}$ and $\mathbf{\Sigma}$. We analyze the case where $\mathbf{\Omega}$ and $\mathbf{\Sigma}$ are simultaneously diagonalizable with eigenvalues $\omega_k$ and $\lambda_k$. In this case, $\mathbf{\Gamma}$ will share the same eigenbasis. Let the eigenvalues be $\gamma_k$

$$\frac{\partial}{\partial t}\gamma_k(t) = 2\omega_k\lambda_k^2 \left( 1 - L^{-1}\lambda_k\gamma_k(t) \right)^{2L-1}. \tag{95}$$

The fixed point of the above dynamics is $\gamma_k = L\lambda_k^{-1}$. In the large depth limit $L \to \infty$, we can solve the dynamics exactly

$$\frac{d}{dt}\gamma_k(t) = 2\omega_k\lambda_k^2 e^{-2\lambda_k\gamma_k(t)} \implies e^{2\lambda_k\gamma_k}d\gamma_k = \omega_k\lambda_k^2 dt$$

$$\implies 2\lambda_k\gamma_k(t) = \ln(1 + 4\omega_k\lambda_k^3 t) \tag{96}$$

Plugging this solution into the loss function we find

$$\mathcal{L}(t) = \frac{1}{D}\sum_k \omega_k\lambda_k e^{-2\gamma_k\lambda_k} = \frac{1}{D}\sum_k \omega_k\lambda_k[1 + \omega_k\lambda_k^3 t]^{-1} \tag{97}$$

Under powerlaw (source/capacity) assumptions on the structure of the data

$$\lambda_k \sim k^{-\nu} , \; \omega_k\lambda_k \sim k^{-\nu\beta-1}, \tag{98}$$

the loss scales in a powerlaw as

$$\mathcal{L}(t) = \sum_k k^{-\nu\beta-1}[1 + k^{-2\nu-\nu\beta}t]^{-1} \sim t^{-\frac{\beta}{\nu+\nu\beta+1}} \tag{99}$$

This powerlaw provides a decent approximation to large but finite depth models $L$ as we show in Figure 3.

## E  RANDOMLY ROTATED AND STRUCTURED COVARIANCE (**RRS**) SETTING

### E.1  EXPLOITING SYMMETRY IN THE GRADIENT UPDATES

Utilizing the definition of the RRS setting, we can massage the placement of the orthogonal matrices so that the loss function can be expressed as

$$\mathcal{L} = \left\langle |\mathbf{\Lambda}^{1/2} \left( \mathbf{I} - L^{-1}\mathbf{O}^\top\mathbf{\Gamma}\mathbf{O}\hat{\mathbf{\Sigma}} \right)^L \boldsymbol{\beta}_\star|^2 \right\rangle. \tag{100}$$

From this expression, it is immediately clear that this function is rotationally invariant with respect to $\mathbf{\Gamma}$ since the transformation $\mathbf{\Gamma} \to \mathbf{V}\mathbf{\Gamma}\mathbf{V}^\top$ for orthogonal $\mathbf{V}$ leaves the Haar average unchanged. The form of the loss gradients also reveals a symmetry

$$\frac{\partial}{\partial \mathbf{\Gamma}}\mathcal{L} = -2L^{-1}\sum_\ell \left\langle \mathbf{O}\hat{\mathbf{\Sigma}}\mathbf{M}^\ell\boldsymbol{\beta}_\star\boldsymbol{\beta}_\star^\top \left( \mathbf{M}^L \right)^\top \mathbf{\Lambda} \left( \mathbf{M}^{L-1-\ell} \right)^\top \mathbf{O}^\top \right\rangle , \; \mathbf{M} = \mathbf{I} - \mathbf{O}^\top\mathbf{\Gamma}\mathbf{O}\hat{\mathbf{\Sigma}} \tag{101}$$

Note that starting from zero initialization, we have $\mathbf{M} = 0$ which implies isotropic gradients $\frac{\partial}{\partial \mathbf{\Gamma}}\mathcal{L}|_{\mathbf{\Gamma}=0} \propto \mathbf{I}$. To see this note that $\left\langle \mathbf{O}\mathbf{C}\mathbf{O}^\top \right\rangle \propto \mathbf{I}$ for any matrix $\mathbf{C}$ that is independent of $\mathbf{O}$. Further, suppose that we evaluated the loss gradient at any isotropic $\mathbf{\Gamma} = \gamma\mathbf{I}$, then $\mathbf{M} = \mathbf{I} - \gamma\hat{\mathbf{\Sigma}}$ and the gradients remain isotropic

$$\frac{\partial \mathcal{L}}{\partial \mathbf{\Gamma}} = -2L^{-1}\sum_\ell \left\langle \mathbf{O}\hat{\mathbf{\Sigma}}\mathbf{M}^\ell\boldsymbol{\beta}_\star\boldsymbol{\beta}_\star^\top \left( \mathbf{M}^L \right)^\top \mathbf{\Lambda} \left( \mathbf{M}^{L-1-\ell} \right)^\top \mathbf{O}^\top \right\rangle \tag{102}$$

$$= -2L^{-1}\sum_\ell \text{tr} \left\langle \hat{\mathbf{\Sigma}}\mathbf{M}^\ell\boldsymbol{\beta}_\star\boldsymbol{\beta}_\star^\top \left( \mathbf{M}^L \right)^\top \mathbf{\Lambda} \left( \mathbf{M}^{L-1-\ell} \right)^\top \right\rangle \times \mathbf{I} \tag{103}$$

where we explicitly performed the average over the only remaining factors of $\boldsymbol{O}$ since $\boldsymbol{M} = \boldsymbol{I} - \gamma\hat{\boldsymbol{\Sigma}}$ is independent of $\boldsymbol{O}$. Further, we can express the dynamics of $\gamma(t)$ as a gradient flow on the reduced one-dimensional loss landscape

$$\mathcal{L}(\gamma) = \operatorname{tr} \boldsymbol{\Omega} \left( \boldsymbol{I} - \gamma L^{-1}\hat{\boldsymbol{\Sigma}} \right)^L \boldsymbol{\Lambda} \left( \boldsymbol{I} - \gamma L^{-1}\hat{\boldsymbol{\Sigma}} \right)^L \tag{104}$$

# F    SCALING LAW THEORY

In this section, we consider the scaling law theory with a finite projection matrix $\boldsymbol{A}$, following Bordelon et al. (2024a). In this case, our **RRS** setting requires utilizing the free product result in Appendix A. However, we will first describe a modification of the two point deterministic equivalent outside of the proportional scaling limit.

## F.1    NON-PROPORTIONAL SCALING LIMITS

To make contact with more realistic scaling law theory, we are interested in varying large but finite $N$ and $P$ for an arbitrary (trace class) spectrum

$$\operatorname{Tr} \boldsymbol{\Lambda} < \infty. \tag{105}$$

This is the setting, for instance, where kernel methods and power-law random feature models have been analyzed successfully Bordelon et al. (2020; 2024a); Defilippis et al. (2024). In this case, the order of limits is to first take $D \to \infty$ at finite (but large) $N$ and $P$. We stress that (unlike the proportional limit), this result gives an approximation rather than an exact description, as traces remain random variables for finite rank $\boldsymbol{A}$ or $\hat{\boldsymbol{\Sigma}}$. However, as demonstrated empirically in Bordelon et al. (2024a) this provides an excellent agreement at even very modest values for $N, P$ on powerlaw features.

We now work out a deterministic equivalent for $\boldsymbol{M} = \boldsymbol{OBO}^\top \boldsymbol{A} \in \mathbb{R}^{D \times D}$ [5]. This requires a slight modification of the two-point deterministic equivalent equations from Appendix A. The action in this case has the form

$$\mathcal{S}[\boldsymbol{\Sigma}, \boldsymbol{\Psi}] = -\operatorname{Tr}\boldsymbol{\Sigma}\boldsymbol{\Psi} - \ln\mathcal{Z}_A(\boldsymbol{\Psi}) - \operatorname{Tr}\boldsymbol{\Sigma}\hat{\boldsymbol{\Sigma}}_\star - \ln\det\left(\hat{\boldsymbol{\Sigma}}_\star \otimes \boldsymbol{I} + \boldsymbol{V} \otimes \boldsymbol{B}\right) + D\ln\det\boldsymbol{\Sigma} \tag{106}$$

The saddle point of this action satisfies the following equations

$$\boldsymbol{\Psi} = \boldsymbol{\Sigma}^{-1} - \hat{\boldsymbol{\Sigma}}_\star \,, \quad \boldsymbol{\Sigma}(\omega, \omega') = \left\langle \begin{bmatrix} \boldsymbol{v}(\omega)\boldsymbol{A}^2\boldsymbol{v}(\omega') & \boldsymbol{v}(\omega)\boldsymbol{A}\boldsymbol{\chi}(\omega') \\ \boldsymbol{\chi}(\omega)\boldsymbol{A}^\top\boldsymbol{v}(\omega') & \boldsymbol{\chi}(\omega) \cdot \boldsymbol{\chi}(\omega') \end{bmatrix} \right\rangle \tag{107}$$

$$\boldsymbol{\Sigma} = \operatorname{Tr}\left[\hat{\boldsymbol{\Sigma}}_\star + \boldsymbol{V} \otimes \boldsymbol{B}\right]^{-1} \tag{108}$$

The off-diagonal blocks give

$$\Sigma_{v\chi}(\omega) = \operatorname{Tr}\boldsymbol{A}\left(i\omega + \Psi_{v\chi}(\omega)\boldsymbol{A}\right)^{-1} \equiv \tau_A(i\omega) \tag{109}$$

$$\Psi_{v\chi}(\omega) = \Sigma_{v\chi}^{-1}(\omega) - \hat{\Sigma}_{v\chi}(\omega) \tag{110}$$

Introducing the notation $\tau_M = \operatorname{Tr}\boldsymbol{M}\left(i\omega_M + \boldsymbol{M}\right)^{-1}$ we have

$$i\omega_A(\tau)i\omega_B(\tau) = \left(\frac{D}{\tau} - 1\right)i\omega \tag{111}$$

We note the saddle point value for $\Psi_{v\chi}(\omega) = \frac{i\omega}{i\omega_A}$ can now be expressed entirely in terms of $i\omega$. Similarly, we can express $\Psi_{\chi\chi}(\omega, \omega')$ in terms of $i\omega, i\omega'$ alone. The key deterministic equivalent is thus the same structure as before

$$\boldsymbol{v}(\omega)\boldsymbol{v}(\omega')^\top = (i\omega + \Psi_{v\chi}(\omega)\boldsymbol{A})^{-1}\left[\boldsymbol{v}_0\boldsymbol{v}_0^\top - \Psi_{\chi\chi}(\omega, \omega')\boldsymbol{I}\right](i\omega' + \Psi_{v\chi}(\omega')\boldsymbol{A})^{-1}. \tag{112}$$

---

[5]In this section, we use $D$ for the dimension of $\boldsymbol{A}$ and $\boldsymbol{B}$ to avoid confusion with the projection dimension (width) $N$.

## F.2 DYNAMICS AT INFINITE $N, L, P$

The gradient flow in the limit of $N, L, P \to \infty$ has the form

$$\frac{d}{dt}\gamma(t) = -\frac{\partial}{\partial\gamma}\sum_k \lambda_k \omega_k e^{-2\gamma(t)\lambda_k} \approx -\frac{\partial}{\partial\gamma}\gamma(t)^{-\beta} = \beta\gamma(t)^{-\beta-1} \tag{113}$$

This is a separable ODE with solution (under the assumption that $\gamma(0) = 0$) up to constants

$$\gamma(t) \sim t^{\frac{1}{\beta+2}}. \tag{114}$$

This implies a powerlaw loss scaling with exponent set by the rate of growth of $\gamma(t)$

$$\mathcal{L}(t) \sim \gamma(t)^{-\beta} \sim t^{-\frac{\beta}{2+\beta}}. \tag{115}$$

We expect these dynamics to hold in the limit of $N, L, P \to \infty$. At finite $N, L, P$ the model will saturate at some finite maximal value of $\gamma$.

## F.3 DEPTH SCALING LAW AT INFINITE TIME, WIDTH, AND CONTEXT LENGTH

We now explore the scaling law in depth $L$. In this case, we can consider taking all other resources to infinity $t, N, P \to \infty$. The value of $\gamma$ will stabilize, resulting in a final loss

$$\mathcal{L}(\gamma) = \sum_k \lambda_k \omega_k \left(1 - \gamma L^{-1}\lambda_k\right)^{2L} \tag{116}$$

While $\gamma(t)$ diverges as $t^{\frac{1}{2+\beta}}$ in the infinite depth limit, we see that at large but finite depth, there is a maximal $\gamma$ which enables stability along the top eigendirection. Specifically we need $\gamma < \frac{2L}{\lambda_1}$ and approximate the optimal value s

$$\gamma_\star \approx L/\lambda_1. \tag{117}$$

where $\lambda_1$ is the top (maximal eigenvalue). At $\gamma_\star$, the error takes the form

$$\mathcal{L} \approx \sum_k \lambda_k (\beta_k^\star)^2 \left(1 - \lambda_k/\lambda_1\right)^L \approx \int dk\, k^{-\nu\beta-1} \exp(-Lk^{-\nu}) \approx L^{-\beta}. \tag{118}$$

which matches the scaling of $L$ steps of gradient descent on a problem with source exponent $\beta$ Bordelon & Pehlevan (2021).

## F.4 MAPPING THE LOSS FUNCTION TO A TWO POINT DETERMINISTIC EQUIVALENT

We utilize dynamical mean field theory (DMFT) result of Appendix A to compute the effective loss as a function of $N, L$ and $\gamma$. We start with a representation of the relevant polynomial in $M$ with the Cauchy integral formula

$$\left(I - \gamma L^{-1}M\right)^L = \int_{\mathcal{C}} \frac{d\omega}{2\pi} \left[i\omega + M\right]^{-1} \left(1 + \gamma L^{-1}i\omega\right)^L, \tag{119}$$

where the contour $\mathcal{C}$ encloses the positive imaginary axis in complex plane (Bender & Orszag, 2013). Next, we define $v(\gamma) = \left(I - \gamma L^{-1}M\right)^L \beta_\star$

$$\mathcal{L} = \langle v(\gamma)^\top \Lambda v(\gamma) \rangle = \operatorname{Tr} \Omega \left[\left(I - \gamma L^{-1}M\right)^L\right]^\top \Lambda \left(I - \gamma L^{-1}M\right)^L \tag{120}$$

The loss can thus be expressed as a double integral involving the resolvents

$$\mathcal{L} = \int_{\mathcal{C}\times\mathcal{C}} \frac{d\omega d\omega'}{(2\pi)^2} (1 + \gamma L^{-1}i\omega)^L (1 + \gamma L^{-1}i\omega')^L \operatorname{Tr} \Omega \left[\left[i\omega + M\right]^{-1}\right]^\top \Lambda \left[i\omega' + M\right]^{-1} \tag{121}$$

The main result needed is the following deterministic equivalent from Appendix A.

$$\operatorname{Tr} \Omega \left[\left[i\omega + M\right]^{-1}\right]^\top \Lambda \left[i\omega' + M\right]^{-1}$$

$$\simeq \operatorname{Tr} \left(i\omega + \Psi_{v\chi}(\omega)A\right)^{-1} \left[\Omega - I\Psi_{\chi\chi}(\omega, \omega')\right] \left(i\omega' + \Psi_{v\chi}(\omega')A\right)^{-1} \Lambda \tag{122}$$

where $\Psi_{v\chi}(\omega) = i\omega/i\omega_{(A^\top A)^2}$ can be determined from the equation

$$i\omega_{(A^\top A)^2}(\tau) i\omega_{\hat{\Sigma}}(\tau) = \frac{1-\tau}{\tau}(i\omega) \tag{123}$$

Once $\tau$ is identified as a function of $\omega$, we can compute $\Psi_{v\chi}$ and $\Psi_{\chi\chi}$ from the formulae in A.

## F.5 ORTHOGONAL PROJECTION AND STRUCTURED GAUSSIAN EMPIRICAL COVARIANCE

In this section, we describe what these relations imply for a product matrix that arises in our **RRS** setting, which is a free product of the width projection matrix $(A^\top A)^2$ where $A \in \mathbb{R}^{N \times D}$ has rank at most $N$ and the empirical covariance for data in the context $\hat{\Sigma} = \frac{1}{P} X X^\top \in \mathbb{R}^{D \times D}$,

$$M = O \left(A^\top A\right)^2 O^\top \hat{\Sigma}, \tag{124}$$

where $O$ is a random orthogonal matrix sampled from the Haar measure. The data $X \in \mathbb{R}^{P \times D}$ are comprised of random iid vectors with covariance $\Lambda$, which we take to be diagonal without loss of generality. Further, we let $A^\top A$ be a rank $N$ projection matrix

$$A^\top A = \sum_{k=1}^{N} e_k e_k^\top \in \mathbb{R}^{D \times D} \tag{125}$$

where $e_k \in \mathbb{R}^D$ are Cartesian (one-hot) unit vectors. To utilize the result in Appendices A and F.1, we first need to compute the necessary resolvents for these two matrices.

$$\tau = \text{Tr}(A^\top A)^2 \left(i\omega_{(A^\top A)^2} + (A^\top A)^2\right)^{-1} = \frac{N}{i\omega_{(A^\top A)^2} + 1} \implies i\omega_{(A^\top A)^2} = -1 + \frac{N}{\tau} \tag{126}$$

The other matrix can be easily worked out for a structured empirical covariance with $P$ samples following the techniques of Bordelon et al. (2024a) which reveals that

$$\tau = \text{Tr}\,\hat{\Sigma} \left(i\omega_{\hat{\Sigma}} + \hat{\Sigma}\right)^{-1} = \text{Tr}\Lambda \left(i\omega_{\hat{\Sigma}}(1 - P^{-1}\tau)^{-1} + \Lambda\right)^{-1} = \tau_\Lambda \left(i\omega_{\hat{\Sigma}}(1 - P^{-1}\tau)^{-1}\right). \tag{127}$$

Using our result for products we find

$$\left(-1 + \frac{N}{\tau}\right)\left(i\omega_{\hat{\Sigma}}(\tau)\right) = \left(-1 + \frac{D}{\tau}\right)i\omega$$
$$\tau = \text{Tr}\Lambda \left(i\omega_{\hat{\Sigma}}(1 - P^{-1}\tau)^{-1} + \Lambda\right)^{-1}$$
$$= \text{Tr}\Lambda \left(i\omega \left(-1 + D\tau^{-1}\right)(-1 + N\tau^{-1})^{-1}(1 - \tau P^{-1})^{-1} + \Lambda\right)^{-1}. \tag{128}$$

This final equation provides $\tau(\omega)$ which can be inverted to derive the final deterministic equivalent as we outline in F.1.

**Width Bottleneck** Now we obtain the scaling of the loss with width $N$ in the regime where $t, P, L, D \to \infty$. To do, we examining the structure of the equations at $P \to \infty$ and $i\omega \to 0$, which correspond to taking gradient flow time $\gamma \to \infty$ (which is the correct solution for noise free problems at infinite depth). In this limit $\tau = N$ so we have that

$$\tau = N = \text{Tr}\Lambda \left(i\omega_{\hat{\Sigma}} + \Lambda\right)^{-1}. \tag{129}$$

For powerlaw features $\lambda_k \sim k^{-\nu}$ we find an approximate solution for $i\omega_{\hat{\Sigma}}$

$$\lambda_k \sim k^{-\nu} \implies i\omega_{\hat{\Sigma}} \approx N^{-\nu}. \tag{130}$$

Thus the loss at width $N$ and context length $P$ can be approximated as

$$\lim_{t, L, P \to \infty} \mathcal{L}(t, N, P, L) = \sum_k \frac{(i\omega_{\hat{\Sigma}})^2 \lambda_k (\beta_k^\star)^2}{(i\omega_{\hat{\Sigma}} + \lambda_k)^2} \sim \sum_k \frac{k^{-\nu\beta - 1}}{(1 + k^{-\nu} N^\nu)} \approx N^{-\nu\beta} \tag{131}$$

**Context Bottleneck** Following the same logic, we can investigate the regime where performance is limited by context $P$. To access an approximation for this regime, we take $N \to D$ and $i\omega \to 0$. In this case, we have $i\omega = i\omega_{\hat{\Sigma}}$. Since $1 - \frac{\tau}{P} \to 0$ as $i\omega \to 0$, we introduce the leading order behavior for small $i\omega$

$$1 - \frac{\tau}{P} \sim r^{-1} i\omega \,, \; i\omega \to 0 \tag{132}$$

where $r$ is a (currently) unknown value. Plugging this into the self consistency equation, we identify the value of $r$

$$P = \text{Tr}\mathbf{\Lambda}[r + \mathbf{\Lambda}]^{-1} = \sum_k \frac{\lambda_k}{r + \lambda_k} \approx r^{-\frac{1}{\nu}} \tag{133}$$

which gives the scaling $r \approx P^{-\nu}$. We can now use the final value of the resolvent $\lim_{\omega \to 0} i\omega \, (i\omega + \mathbf{\Lambda})^{-1} = (r + \mathbf{\Lambda})^{-1}$

$$\mathcal{L} = \sum_k \frac{r^2 \lambda_k (\beta_k^\star)^2}{(r + \lambda_k)^2} \approx \sum_k \frac{k^{-\nu\beta-1}}{(1 + k^{-\nu} P^\nu)^2} \sim P^{-\nu\beta}. \tag{134}$$

**Rank Deficiency/Null Space Interpretation** These last two scaling laws express the simple fact that rank $N$ or rank $P$ matrices only allow the top $N$ or $P$ eigendirections to be learned (Bordelon et al., 2024a).

## G    ENHANCING REALISM OF THE MODEL

### G.1    DIFFERENT PARAMETERS FOR EACH LAYER

In this section, we analyze the case where the model is allowed $L$ distinct attention layers along the residual stream $\{\mathbf{\Gamma}^\ell\}_{\ell=1}^L$ instead of a single attention layer matrix $\mathbf{\Gamma}$ which is applied $L$ times recurrently. We focus our attention on the RRS setting at large width and large context length where the model still exhibits nontrivial depth and pretraining time scaling laws. We first consider the noise free setting $\sigma^2 = 0$ before considering the role of target noise.

### G.1.1    RRS COVARIATES $\to$ ISOTROPIC $\mathbf{\Gamma}$ $\to$ LAYERS REMAIN IDENTICAL

First, we note that by the rotational invariance, each of the matrices is isotropic $\mathbf{\Gamma}^\ell = \gamma^\ell \mathbf{I}$ if they are initialized as isotropic (such as zero initialization). As a consequence, the predictor on dataset $\mathbf{X}, \mathbf{y}$ and test point $\mathbf{x}_\star$ has the form

$$f(\mathbf{x}_\star) = \frac{1}{LP} \mathbf{x}_\star^\top \mathbf{X} \sum_{\ell=1}^L \gamma_\ell \prod_{k=0}^{\ell-1} \left( \mathbf{I} - \frac{1}{LP} \gamma_k \mathbf{X}^\top \mathbf{X} \right) \mathbf{y} \tag{135}$$

We note through a simple inductive argument, that the matrix product can be rearranged into a form that is clearly **permutation symmetric** in the variables $\{\gamma_\ell\}_{\ell=1}^L$

$$\sum_{\ell=1}^L \gamma_\ell \prod_{k=0}^{\ell-1} \left( \mathbf{I} - \frac{1}{LP} \gamma_k \mathbf{X}^\top \mathbf{X} \right) = \sum_{n=1}^L (-1)^{n-1} \left[ \frac{1}{LP} \mathbf{X}^\top \mathbf{X} \right]^{n-1} \sum_{k_1 \neq k_2 ... \neq k_n} \gamma_{k_1}...\gamma_{k_n} \tag{136}$$

Thus because the predictor is permutation symmetric, the loss function is also permutation symmetric in the $\{\gamma^\ell\}$ variables. We therefore expect permutation symmetric dynamics and a solution where $\gamma^\ell = \gamma$ provided the initial condition is symmetric $\gamma^\ell(0) = \gamma(0)$. Indeed, analyzing the gradient flow from $\gamma^\ell = 0$ for all $\ell$ leads to a balanced solution where all of these are equal since

$$\frac{\partial}{\partial \gamma^\ell} \mathcal{L}(\{\gamma^\ell\})|_{\mathbf{\gamma}=\gamma\mathbf{1}} = \frac{\partial}{\partial \gamma^{\ell'}} \mathcal{L}(\{\gamma^\ell\})|_{\mathbf{\gamma}=\gamma\mathbf{1}} \,, \; \forall \ell, \ell' \in \{1,...,L\}. \tag{137}$$

Thus gradient flow will maintain a balance in these parameters. Further, this flow will exactly match the dynamics of the recurrent model if the learning rate is upscaled by $L$. We plot these dynamics and show balancing of the $\gamma^\ell(t)$ variables in Figure 8.

### G.2    DECOUPLING WEIGHTS IN ATTENTION LAYERS

We note that for the **ISO** and **RRS** settings, we can exploit similar symmetry arguments as in C.2 and E to argue that the gradient updates for $\mathbf{W}_x, \mathbf{W}_k, \mathbf{W}_q$ are isotropic and that $\mathbf{W}_v$ gets an update in the $\mathbf{w}_o \mathbf{w}_y^\top$ direction. The gradient flow can be expressed in terms of the original gradients on $\mathbf{\Gamma}$

$$\partial_t \mathbf{W}_i = -\frac{\partial \mathcal{L}}{\partial \mathbf{\Gamma}} \cdot \frac{\partial \mathbf{\Gamma}}{\partial \mathbf{W}_i} \,, \; i \in \{x, k, q, v, o, y\} \tag{138}$$

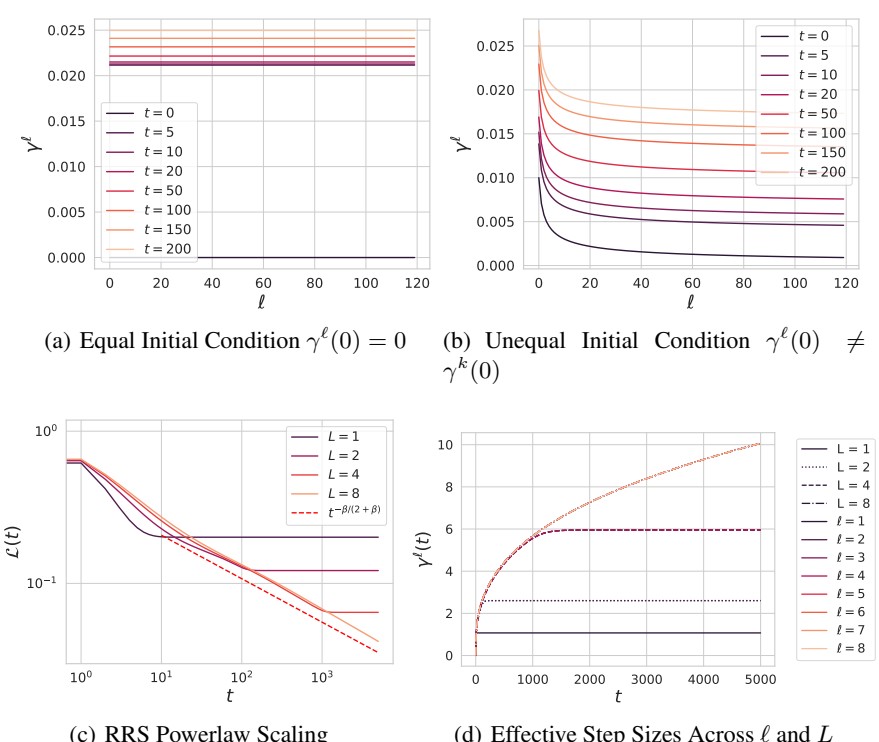

(a) Equal Initial Condition $\gamma^\ell(0) = 0$

(b) Unequal Initial Condition $\gamma^\ell(0) \neq \gamma^k(0)$

(c) RRS Powerlaw Scaling

(d) Effective Step Sizes Across $\ell$ and $L$

Figure 8: Pretraining on population gradient flow with **RRS** covariates under decoupled layers $\{\mathbf{\Gamma}^\ell\}$ is equivalent to coupled layers ($\mathbf{\Gamma}^\ell = \mathbf{\Gamma}$) if the initial condition is symmetric. We plot $\gamma^\ell$ for a depth $L = 120$ trained on $D = 120$ dimensions with $P = 100$ and $\sigma = 0.5$. (a) If the ICL model is initialized with all $\gamma^\ell = \gamma$ and pretrained with **RRS** covariates, then the symmetry is maintained throughout pretraining. (b) In the absence of symmetry in the initial condition, there is not symmetry in the final configuration. (c) Dynamics for layer-decoupled reduced $\Gamma$ model. The loss dynamics under powerlaw RRS covariates exhibit the same powerlaws. (d) The different scale factors remain balanced throughout gradient flow $\gamma^k(t) = \gamma(t)$ for all $k \in \{1, ..., L\}$ due to permutation symmetry.

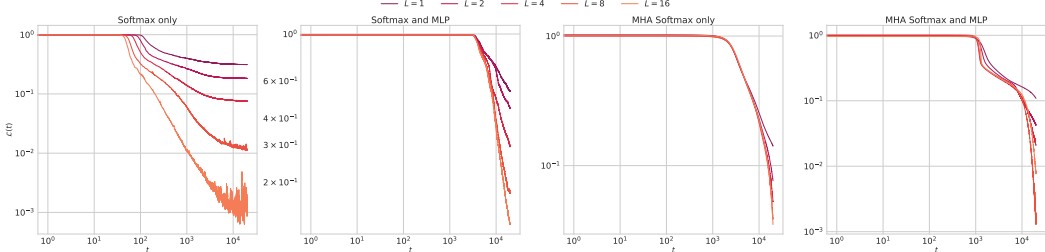

Figure 10: Varying depth in softmax, softmax+MLP, multihead attention and all three. Softmax attention networks exhibit clear depth separation. The dynamics become more consistent across depths for increasing number of heads with softmax attention.

As before, under the assumption of small initial conditions, we can reduce the loss to a collection of ODEs on scalars representing the scale of each weight matrix

$$\mathcal{L}(w_o, w_v, w_q, w_k, w_x, w_y) = \text{tr} \, \mathbf{\Omega} \mathbf{\Lambda} \left[ \mathbf{I} - w_o w_y \left( \mathbf{I} - \left[ 1 - L^{-1} w_v w_k w_q (w_x)^2 \mathbf{\Lambda} \right]^L \right) \right]^2 \quad (139)$$

Gradient flow dynamics on this loss function can reproduce the dynamics of the decoupled self attention model. Under the further assumption that $w_o = w_y = 1$, we can simplify the loss further to

$$\mathcal{L}(w_v, w_q, w_k, w_x) = \text{tr} \, \mathbf{\Omega} \mathbf{\Lambda} \left[ \mathbf{I} - L^{-1} w_v w_k w_q (w_x)^2 \mathbf{\Lambda} \right]^{2L} \quad (140)$$

We note that $w_x$ will be updated twice as quickly as the other weights under gradient flow. To achieve balance, we can initialize it to have $w_x(0) = \sqrt{2} w_k(0)$ and have $w_q(0) = w_k(0) = w_v(0) = \sigma$. In this case, the loss can be further reduced to a function of a single variable

$$\mathcal{L}(w_v, w_q, w_k, w_x) = \text{tr} \, \mathbf{\Omega} \mathbf{\Lambda} \left[ \mathbf{I} - L^{-1} w(t)^5 \mathbf{\Lambda} \right]^{2L} \quad (141)$$

Under source and capacity assumptions, this will generate the following dynamics at large depth $L$

$$w(t) = t^{\frac{5}{5\beta+2}} \implies \mathcal{L} \sim t^{-\frac{5\beta}{5\beta+2}}. \quad (142)$$

### G.3 SOFTMAX ATTENTION

We also provide numerical experiments with non-recurrent softmax attention, decoupled layers, and Adam. In this context we use CompleteP scaling for the learning rate $\Theta_L(1)$ (Dey et al., 2025). The results are provided in Figure 6 (c).

Supplementing this figure is an equivalent figure for a non-recurrent architecture with alternating softmax attention and MLP with GELU activation, given by Figure 9.

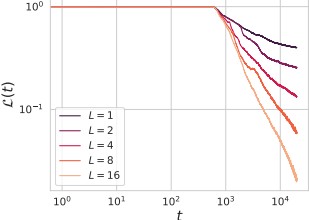

Figure 9: Softmax attention model with MLP Blocks on the residual stream also benefit from increasing the depth.

