# OpenReview forum: "Theory of Scaling Laws for In-Context Regression: Depth, Width, Context and Time"
_ICLR.cc/2026/Conference — ICLR 2026 Poster_

### Official Review · Reviewer_NdPw · 2025-10-27

**Soundness:** 4
**Presentation:** 2
**Contribution:** 4
**Rating:** 6
**Confidence:** 4

**Summary:**

This paper theoretically investigates in-context learning for linear regression using deep linear self-attention models, analyzing performance based on width (N), depth (L), context length (P), pretraining time (t), and data structure. Examining isotropic (ISO), fixed structured (FS), and randomly rotated structured (RRS) covariance settings, the authors find that depth primarily benefits ICL in the ISO and FS settings only when context length is limited; for infinite context length in these settings, increasing depth beyond L=1 offers no advantage. However, in the more complex RRS setting where covariances vary, depth significantly improves performance even at infinite context length. For this RRS case, the paper derives a Chinchilla-like scaling law and predicts a compute-optimal shape scaling, linking optimal architecture to task data properties.

**Strengths:**

- The paper provides a comprehensive theoretical analysis of multi-layer linear self-attention models for in-context linear regression across three distinct covariate settings (ISO, FS, RRS).
- This paper rigorously characterizes the training dynamics using gradient flow analysis, revealing how the model learns under different data structures and providing an interpretation of the learned estimator as implementing multi-step gradient descent with optimal step sizes.
- The derivation of a Chinchilla-like neural scaling law incorporating time, width, depth, and context length for the RRS setting in the context of linear regression with power-law features is a significant theoretical contribution.
- The application of Dynamical Mean Field Theory (DMFT) to derive a two-point deterministic equivalent for the loss landscape under random rotations represents a novel technical approach for analyzing complex learning dynamics in this asymptotic regime.

**Weaknesses:**

-The presentation of detailed proofs and derivations within the appendix could be improved for clarity and accessibility, making it challenging to fully verify the technical steps.

**Questions:**

- Could the authors elaborate on the necessity of employing DMFT to derive the closed-form loss expression in Result 7? Is this approach required because directly analyzing the gradient flow dynamics in (13) is intractable, perhaps due to the lack of a known closed-form solution for the ODE governing $\gamma(t)$ in the randomly rotated setting with finite width N?
- The current analysis focuses heavily on the proportional asymptotic regime. Are the techniques employed amenable to deriving non-asymptotic results that might provide insights into the behavior of the system with sizes?
- The derivation of the two-point deterministic equivalent using DMFT in Appendix A introduces notation that appears distinct from the parameters used in the main text to describe the Transformer model and its dynamics and it's hard to directly match (22) with the main result. Could the authors provide a clearer mapping between the DMFT variables/order parameters and the model parameters/dynamics described earlier in the paper?

---

> ### Author Response · Authors · 2025-11-20
>
> We thank the reviewer for their feedback. We have tried to improve the exposition around our theoretical results and expand Appendix A to include more introduction to DMFT as a method. We hope in light of these updates the reviewer will consider increasing their score.
>
> *-The presentation of detailed proofs and derivations within the appendix could be improved for clarity and accessibility, making it challenging to fully verify the technical steps.*
>
> We have made an effort to improve the clarity and accessibility of the calculations. We added two new subsections in Appendix A, **Intuitive Motivation for this Section** and **Dynamical Mean Field Theory Intuitive Example**. The former discusses connections between the calculation in this section
>
>
> *Could the authors elaborate on the necessity of employing DMFT to derive the closed-form loss expression in Result 7?*
>
> Yes, the key thing is that the dynamical system governing the forward pass involve a linear dynamical system driven by an *asymmetric random matrix*
>
> $$M = O (A^\top A)^2 O^\top \hat{\Sigma}$$
>
> with $v^{\ell+1} = (I - L^{-1} \gamma M) v^\ell$. We then want to compute the correlation matrix for $\left<v^L v^L \right>$. DMFT is a tool which can easily handle high dimensional averages over this kind of disordered dynamics. Specifically, it provides access to two-point (correlation functions)
>
> $$\left< [i\omega + M]^{-1} \Omega ([i\omega' + M]^{-1})^\top \right>$$
>
> for arbitrary matrices $\Omega$ as we describe in the new version of Appendix A in the subsection titled **Intuitive Motivation for this Section** and the next section **Dynamical Mean Field Theory Intuitive Example**.
>
>
> We added the following sentence before stating Result 7:
>
> **As the matrix $M = O (A^\top A )^2 O^\top \hat{\Sigma}$ driving the dynamics is asymmetric, we characterize the loss landscape using dynamical mean field theory, a technique from the physics of spin glasses which allows asymptotic descriptions of high dimensional disordered dynamical systems.**
>
>
> *The current analysis focuses heavily on the proportional asymptotic regime. Are the techniques employed amenable to deriving non-asymptotic results that might provide insights into the behavior of the system with sizes?*
>
> Yes, we now focus on a version of our theory for non-proportional scalings. We added Appendix F.1 and updated the rest of Appendix F and the formula in Result 7.
>
>
> *The derivation of the two-point deterministic equivalent using DMFT in Appendix A introduces notation that appears distinct from the parameters used in the main text to describe the Transformer model and its dynamics and it's hard to directly match (22) with the main result. Could the authors provide a clearer mapping between the DMFT variables/order parameters and the model parameters/dynamics described earlier in the paper?*
>
> Yes, we now provide an additional subsection in Appendix A called **Intuitive Motivation for this Section** which maps the ICL error to this computation. In this section, we show how the ICL loss can be directly obtained from the two point result we derived.

---

### Official Review · Reviewer_qeR8 · 2025-10-30

**Soundness:** 2
**Presentation:** 1
**Contribution:** 2
**Rating:** 2
**Confidence:** 3

**Summary:**

This paper studies deep linear self-attention trained on the in-context linear regression task, characterizing how performance depends on width, depth, number of training steps, batch size and data per context.

**Strengths:**

The paper studies an interesting and relevant problem of training a deep linear self-attention model on in-context linear regression tasks, considering different in-context task structures. Extensive prior work has examined one-layer linear self-attention trained on in-context linear regression with isotropic task vectors. Analyzing deep models and non-isotropic task vectors represents meaningful progress.

**Weaknesses:**

- My main concern is the looped transformer assumption, $W_i^l=W_i^{l'}, i\in \{k,q,v\}$, which means that all layers have the same weights. The expressivity and optimization properties of a deep self-attention model can differ significantly with or without this assumption. Why would the scaling limits derived under this constrain reflect those of a real multi-layer attention model, which typically learns different weights across layers?

- In the reduced model in Equation (4), all weight matrices within a layer appear to be merged into a single trainable matrix $\Gamma$, effectively making the self-attention layer "shallow". Since gradient descent dynamics and the loss landscape are sensitive to such reparameterization, the true loss landscape probably differs from those shown in Figure 2 and Figure 4b. If this is indeed the case, it would be helpful to explicitly highlight this distinction.

- I suggest that the authors perform another round of proofreading and polishing. There are presentation issues that make the paper unnecessarily difficult to read smoothly. I list some below.

  It appears that manual vertical spacing commands have been used in several places of the paper. The formatting on page 14 seems irregular.

  The clarity of Equation (3) could be improved by specifying the dimensionality of weight matrices.

  The authors use inner-product and transpose notations interchangeably. The expectation notation $\mathbb E(\cdot)$ and the angle brackets notation $\langle \cdot \rangle$ are also used interchangeably. Adopting consistent notations throughout would enhance readability.

  The symbols $\boldsymbol X, \boldsymbol y$ in Equation (4) seem to be undefined.

  The symbol $i$ is used inconsistently: sometimes as an index and sometimes as the imaginary number. In particular, it is undefined in Result 7. Clarifying its meaning in each context would avoid confusion.

  In Equations (13), (15), it appears that a scalar is being added to a matrix, e.g., $(1-L^{-1}\gamma\Lambda), (i\omega+\Psi\Lambda)$. Please check these terms for dimensional consistency. Additionally, the trace operator $\text{tr}$ is used without parentheses, which could be ambiguous.

**Questions:**

The problem considered in this paper is interesting and potentially important. However, recurring issues with presentation and clarity make it difficult to fully assess the contributions. Improving the clarity would make the results more accessible for proper evaluation.

---

> ### Author Response · Authors · 2025-11-20
>
> We thank the reviewer for their feedback. We have tried to address their main concerns and would appreciate a reevaluation of the work in light of these updates.
>
> ### Weaknesses
>
> *My main concern is the looped transformer assumption, which means that all layers have the same weights. The expressivity and optimization properties of a deep self-attention model can differ significantly with or without this assumption. Why would the scaling limits derived under this constrain reflect those of a real multi-layer attention model, which typically learns different weights across layers?*
>
>
> The new version of our draft **directly addresses this question** in the new Section 5.1 (see new Result 9) with supporting calculations the new Appendix G.1 and simulations in Figure 8. We now prove that **our looped transformer theory** exactly describes the dynamics of a deep model when training on RRS covariates. We verify this empirically in the new Appendix Figure 8.
>
>
> *In the reduced model in Equation (4), all weight matrices within a layer appear to be merged into a single trainable matrix , effectively making the self-attention layer "shallow". Since gradient descent dynamics and the loss landscape are sensitive to such reparameterization, the true loss landscape probably differs from those shown in Figure 2 and Figure 4b. If this is indeed the case, it would be helpful to explicitly highlight this distinction.*
>
> This is a good question. You are correct that the dynamics are distinct, however we show that the change is simple under certain simple assumptions in the RRS setting one can simply reparameterize $\gamma \to w^5$ where $w$ is a scalar since the weights will balance in scale under SGD. We provide a detailed analysis of this in Section 5.2. We added some phrasing emphasizing the difference between the original parameterization and training the $\gamma$ directly. The time exponent under gradient flow can indeed change for powerlaw features as we mention in that section.
>
> *I suggest that the authors perform another round of proofreading and polishing. There are presentation issues that make the paper unnecessarily difficult to read smoothly. I list some below.*
>
> We thank the reviewer for pointing out all of these writing and notational issues. We have improve these proofreading and polishing issues. We also attempted to make our notation more consistent throughout.
>
> We hope that in light of the updates we made to the paper, that the reviewer would consider increasing their score.

---

### Official Review · Reviewer_B4Ac · 2025-10-31

**Soundness:** 4
**Presentation:** 3
**Contribution:** 3
**Rating:** 8
**Confidence:** 3

**Summary:**

The authors tackle the problem of how should we allocate depth, width, context length, and training compute when scaling transformers for in-context learning (ICL) on regression tasks. The paper analyzes a deep loop linear attention transformer trained (via SGD) to do linear regression in context, without finetuning at test time. It studies three task regimes: (i) isotropic data, (ii) fixed but structured covariance, and (iii) randomly rotated structured covariance (task distribution shifts every context), and derives dynamical equations and asymptotic scaling laws linking pretraining time, width, depth, and context length. The contribution lies in a unified, theoretically grounded scaling law that distinguishes the roles of depth, width, context length, and training time in ICL.

**Strengths:**

1. I appreciate that the paper gives a concrete, theoretically grounded answer to a question that is widely discussed in practice: how should depth vs. width vs. context length scale for in-context learning? Instead of treating “bigger model = better,” it isolates when depth specifically matters, and ties that to properties of the task distribution (shared vs. varying covariance across contexts). Specifically, the model decouples network width $N$ from the problem dimension $D$ when studying the scaling law, and the use of loop transformer ensures that the total number of parameters does not go up with the number of computes---which I believe provides a decoupled and generic test bench that adds to similar work studying scaling with linear model.

2. A demonstration that the usefulness of depth is task-distribution dependent: If all tasks share the same covariance, depth is asymptotically unnecessary (long enough context suffices, as the model weights can "encode" the covariate information). If covariances vary across contexts, depth is fundamentally valuable, even with infinite context (reflecting the philosophy of test-time compute).

3. A good match between theories and experiments.

**Weaknesses:**

1. The entire analysis is built around linear regression tasks solved via in-context learning with (mostly) loop linear attention. I didn't find much discussion surrounding the use of loop attention block. One benefit I could imagine is the decoupling between total model weights and the depth. However, the use of loop could potentially restrict the model's expressiveness, where model could possibly implemented higher-order optimization algorithm, e.g., Newton's step rather than gradient descent [1], and the model might demonstrate different scaling behavior, especially in depth.

2. The key conceptual result is that depth becomes essential when task covariances vary across contexts (“randomly rotated structured,” RRS). But the diversity they model is very specific: random orthogonal rotations of a shared spectrum. That’s mathematically nice but arguably still a stylized shift. Real heterogeneity looks more like mixture of domains, sparsity structure, nonstationary label noise, hierarchical latent factors, etc. It’s not obvious that random rotations is the right stand-in for natural distribution shift.

3. The paper argues its results are relevant to large-scale LLM design, but the experiments cap out at synthetic regression and relatively small controlled transformers. There’s no ablation on modern-scale architectures (residual blocks with MLPs, nonlinear attention heads, long-context finetuning) to show even qualitative alignment. So the significance for frontier models is still somewhat speculative.

[1] Giannou, A., Yang, L., Wang, T., Papailiopoulos, D., & Lee, J. D. (2024). How Well Can Transformers Emulate In-context Newton’s Method? arXiv preprint arXiv:2403.03183.

**Questions:**

1. Your theory and experiments focus on linear regression tasks with (mostly) linear attention, Gaussian feature distributions, and controlled covariance structure. How confident should we be that the same depth–width scaling conclusions hold for nonlinear Transformers trained on natural language, vision, or multimodal data?

2. You mostly analyze single-head linear attention plus residual depth. How do you expect multi-head structure and MLP blocks (i.e., actual transformer blocks) to affect the depth vs. width story? For example, could significantly more attention heads (possibly also scales with D) also benefit the learning process?

---

> ### Author Response · Authors · 2025-11-20
>
> We thank the reviewer for their detailed review, feedback and support of this paper. Below we try addressing the weaknesses and questions.
>
> ### Weaknesses
>
> *The entire analysis is built around linear regression tasks solved via in-context learning with (mostly) loop linear attention. I didn't find much discussion surrounding the use of loop attention block. One benefit I could imagine is the decoupling between total model weights and the depth. However, the use of loop could potentially restrict the model's expressiveness, where model could possibly implemented higher-order optimization algorithm, e.g., Newton's step rather than gradient descent [1], and the model might demonstrate different scaling behavior, especially in depth.*
>
> We thank the reviewer for this question. We have subsequently demonstrated that the looped transformer description is also an exact description of the training for uncoupled transformers for the RRS setting (see new Result 9) and Appendix G.1 and Appendix Figure 8.
>
>
> *The key conceptual result is that depth becomes essential when task covariances vary across contexts (“randomly rotated structured,” RRS). But the diversity they model is very specific: random orthogonal rotations of a shared spectrum. That’s mathematically nice but arguably still a stylized shift. Real heterogeneity looks more like mixture of domains, sparsity structure, nonstationary label noise, hierarchical latent factors, etc. It’s not obvious that random rotations is the right stand-in for natural distribution shift.*
>
> This is a good point. We do not claim that RRS is the most general ICL ensemble but rather is the minimal dataset that generates a depth separation result in this problem.
>
>
> *The paper argues its results are relevant to large-scale LLM design, but the experiments cap out at synthetic regression and relatively small controlled transformers. There’s no ablation on modern-scale architectures (residual blocks with MLPs, nonlinear attention heads, long-context finetuning) to show even qualitative alignment. So the significance for frontier models is still somewhat speculative.*
>
> We are not necessarily claiming that our results or analysis apply to frontier models since the ICL task we consider is quite simple and we focus on simplifications of the architecture for analytical tractability. Rather, we primarily wanted to identify a solveable model where the relative benefits of depth and width can be understood. A motivating plot for us Figure 4 (a-c) in https://arxiv.org/pdf/2505.01618 which shows the existence of an optimal width/depth ratio in deep LLMs trained on next token prediction. We are proposing our toy model as a simple example where depth and width need to be scaled jointly to obtain compute optimal performance.
>
> However, based on this feedback, we are currently running additional experiments on our ICL task including
> 1. Adding MLP blocks
> 2. Varying the number of attention heads
> 3. Comparing softmax and linear attention
>
> In each of these settings, the depth scaling behaves similarly (though not qualitatively exact) for ISO and RRS covariates.
>
> #### Questions
>
> *Your theory and experiments focus on linear regression tasks with (mostly) linear attention, Gaussian feature distributions, and controlled covariance structure. How confident should we be that the same depth–width scaling conclusions hold for nonlinear Transformers trained on natural language, vision, or multimodal data?*
>
> This is a good point and there are many open questions that remain about the connection between this current theory and more realistic data distributions.
>
> *You mostly analyze single-head linear attention plus residual depth. How do you expect multi-head structure and MLP blocks (i.e., actual transformer blocks) to affect the depth vs. width story? For example, could significantly more attention heads (possibly also scales with D) also benefit the learning process?*
>
> The multi-head question is interesting. We have not worked this setting out analytically yet, so we are not entirely sure whether it alters the structure of the learned solution. However, if hazarding a guess based on the analysis of the single head example, under RRS covariates in a linear attention model we expect each head's matrix $\Gamma^\ell_{h} \propto W_x^\top W_k^\top W_q W_x$ would be isotropic due to rotational symmetry of the gradients and the scaling law would be preserved.
>
> We are currently running linear attention experiments with varying heads to examine the solution obtained.

---

### Official Review · Reviewer_KSDs · 2025-10-31

**Soundness:** 3
**Presentation:** 4
**Contribution:** 2
**Rating:** 6
**Confidence:** 4

**Summary:**

This paper studies deep linear self-attention by analyzing the corresponding solvable model. The authors systematically investigate how depth of the model, width, context length, and training steps affect the solution of the model. Specifically, the authors focus on three distinct types of data (termed as ISO, FS, and RRS), where the authors reveal that depth is unnecessary for long contexts on ISO and FS, along with a series of other results that characterize the gradient flow dynamics. Furthermore, the authors derive a separable scaling law for the RSS setting.

**Strengths:**

1. This paper is well written and well organized. The presentation is very clear and the flow of this paper is consistent, which I think can allow the readers to easily appreciate the core contributions of this work (the summarized theoretical results along with immediate numerical experimental results).

   The proof is also easy to follow (though I did not check all the details): first transforming the gradient learning dynamics to an equivalent linear model, which has a simpler dynamics, then diving to this new dynamics under different conditions of the covariance. Although this general idea is not new, incorporating the depth to the analysis is novel.

2. The derived theoretical results indeed consider attention in the multi-layer case, which I believe is an improvement over prior works. The authors demonstrate when and why the depth can be necessary. The summarized results are indeed novel and interesting.

**Weaknesses:**

While the results are interesting by considering the depth, I think the setting is still not significantly novel compared to prior works. In particular:

 - While equation (3) indicates a dependence on the layer depth $l$, the induced parameter $\Gamma$ in fact does not depend on $l$, because the matrices $W_i$'s are treated equally for different layers given one specific $i$. Instead, this dependence on $l$ is replaced by a simple summation over $l$ in $\Gamma$. As a result, the corresponding analysis in fact does not provide significantly novel analysis compared to prior works in this line of research, i.e., studying the gradient flow dynamics of the induced parameter $\Gamma$ (which is still a linear regression) as a proxy of the true weight matrices of the attention model. It remains unclear whether doing so can really capture the effects of depth.

- The RSS is positioned as the most general case, but the randomly rotated and structured across contexts covariance still cannot effectively capture  the essence of task diversity. In addition, the theoretical framework is built on taking a very specific join limit where $P, K, B, D, N \to \infty$ with fixed ratios. While convenient for analytical tractability, this obscures efects that are relevant at finite scales.

- Due to the aforementioned limitations, the generality of the derived results remains unclear.


Minor: As this paper considers solvable models and scaling laws of attention, I think the related work [1], which also studies a solvable model for attention and its scaling laws, could be discussed a bit in the related work.

[1]. Lyu et al.  A Solvable Attention for Neural Scaling Laws. ICLR 2025.

**Questions:**

1. As I'm mostly concerned about the role of the depth, which plays an important role in the novelty of this work, can the authors justify the validity of assuming equal weight matrices across layers and the corresponding generality?

2. Furthermore, can the authors discuss the difficulty brought by varying weight matrices w.r.t layers and how the current framework can still be applied in that case?

3. Is taking the joint limit of $P, K, B, D, N$ necessary for the results presented in this paper?

---

> ### Author Response · Authors · 2025-11-20
> **Response**
>
> We thank the reviewer for their detailed reading and feedback. Below we provide answers to the main concerns and questions raised in the review. We hope that with the new analysis of the decoupled layer case and the new non-proportional scaling analysis, the reviewer will consider increasing their score.
>
> ### Weaknesses
>
> *I think the setting is still not significantly novel compared to prior works*
>
> We disagree with this assessment for a number of reasons.
>
> 1. Our work shows that depth can be generically helpful for ICL regression problems, but is only necessary (at large context sizes) for RRS ICL covariates.
> 2. We clarify that depth is only useful at infinite context length $P \to \infty$ if there is sufficient covariance diversity. We propose the RRS setting as a simple solveable example of a diverse covariance distribution.
> 3. We describe a novel the exact asymptotics of pretraining for random data with context size $P$, width $N$ and depth $L$.
>
> We thus see our work as providing novel advances in both ICL theory and identifying conditions where ICL regression induces a nontrivial depth scaling law.
>
> *The RSS is positioned as the most general case, but the randomly rotated and structured across contexts covariance still cannot effectively capture the essence of task diversity. In addition, the theoretical framework is built on taking a very specific join limit where with fixed ratios. While convenient for analytical tractability, this obscures efects that are relevant at finite scales.*
>
> First, we think that our theory can account for task diversity by varying the rank of the task-correlation matrix $\Omega= \left< \beta \beta^\top \right>$. Suppose that there are $T_{\beta}$ distinct task vectors, then $\Omega$ has rank at most $T_{\beta}$ which could limit the rank of the learned $\Gamma$ matrix. We agree that analysis of distribution shift in the task covariance matrix would be very interesting to explore in future work, but our focus in this work was to generate a depth separation which requires the covariance diversity across contexts.
>
> We agree that the RRS setting is not the most general ICL ensemble. Rather, it is a tractable example that has a depth separation result in the limit as $P \to \infty$. We added a discussion sentence in the Limitations/Future Directions section describing how our setting is not the most general ensemble and some future relaxations of our model.
>
> *In addition, the theoretical framework is built on taking a very specific join limit where $P,K,N,D \to \infty$ with fixed ratios.*
>
> Our theory does not strictly require a joint asymptotic limit. However our theory is **exact** in this limit, and this kind of limit is necessary to describe the isotropic feature case. For large but finite $P,K,N,D$ the theory can still apply but is approximate.
>
> We have added a section in the Appendix and a sentence about a *dimension free* version of our theory in the powerlaw setting when $P,K,N,D$ are all large but finite. In this case, the theory gives a quantitative prediction for the *average/typical loss*, but the loss itself is a random variable that fluctuates at small $N,P$. A similar distinction arises in https://arxiv.org/abs/2402.01092.
>
>
> *As this paper considers solvable models and scaling laws of attention, I think the related work [1], which also studies a solvable model for attention and its scaling laws, could be discussed a bit in the related work*
>
> Thank you for linking to the highly relevant work of Lyu et al which analyzes context length and training time scaling laws for an ICL task in depth 1 attention models. We have added a reference to this work and comparison to our results in the related works section.
>
> *As I'm mostly concerned about the role of the depth, which plays an important role in the novelty of this work, can the authors justify the validity of assuming equal weight matrices across layers and the corresponding generality?*
>
> Yes, we can demonstrate that for ISO or RRS settings, there is an exact equivalence between the dynamics we derived under the equal weight assumption and the dynamics arising from a model with decoupled layers provided that the initialization is isotropic and equal across layers (for instance zero init).
>
> These results are summarized in the new section 5.1 and Result 9 with empirical validation in Figure 8 in the Appendix.
>
>
> *Furthermore, can the authors discuss the difficulty brought by varying weight matrices w.r.t layers and how the current framework can still be applied in that case?*
>
> We were initially motivated by the simplicity of tracking a single matrix to summarize the entire network, but now realize that this equivalence can be argued for ISO and RRS settings.
>
>
> *Is taking the joint limit of $P,N,K,D\to\infty$ necessary for the results presented in this paper?*
>
> No, this is not necessary. We clarify this in the new Appendix F.1 and use general expressions that involve $P,N,D$ in our scaling law section.

---

> > ### Comment · Reviewer_KSDs · 2025-11-21
> >
> > Thank you for your detailed response, which addressed my questions. I have increased my score.

---

### Author Response · Authors · 2025-11-20
**Global Response**

We thank the reviewers for their careful reading and constructive feedback of this work. We have made many improvements to the paper in response to the reviews and would appreciate the reviewers to consider improving their evaluation of the paper. Below we outline some common concerns raised in many of the reviews and what steps we've taken to address them.

### Common Concerns

We now discuss some common concerns which appeared in many of the reviews and the steps we have taken to address them in the new draft. We have made significant updates to the paper to address these concerns and hope that the reviewers will consider improving their scores in light of these updates.

#### Looped transformer vs untied weights

Many reviewers were concerned that we focused our analysis on a model with tied weights across layers ($\Gamma^\ell = \Gamma$ for all $\ell$) instead of having distinct parameters for each attention block.

The new version of our draft **directly addresses this question** in the new Section 5.1 (see new Result 9) with supporting calculations the new Appendix G.1 and simulations in Figure 8.

The key result is that for **ISO** or **RRS** ICL covariates and an initial condition that is symmetric and isotropic across layers $\Gamma^\ell(0) = \gamma(0) I$, there is **no difference** between the dynamics and scaling law that we derived for the looped transformer and the decoupled depth $L$ model. A consequence of this result is that **our looped transformer theory** exactly describes the dynamics of a deep model when training on RRS covariates. We verify this empirically in the new Appendix Figure 8.

#### RRS setting is not the most general ICL data distribution

Some reviewers pointed out that our focus on the RRS setting does not cover the most general possible ICL distribution shift. We appreciate this critique and to respond we would like to emphasize the following points

1. We are primarily interested in showing that **covariance diversity** during pretraining is important to utilize depth to discover an ICL solution that generalizes across data covariances. The RRS setting is but **one example** of a (especially tractable) data distribution that has this property.
2. The nice thing about the RRS model is that it enables analytical calculations through connections to free probability of random matrices which enables us to extract interesting scaling laws. This is not to say that interesting depth scaling laws could not arise for other ICL distributions with diverse covariances.
3. In general, we can easily allow for varying spectra $\Lambda$ across contexts too, but the key effect needed for depth separation is random rotation.

In response to this critique we added some additional discussion in Limitations and Future Directions.

#### Does our theory only hold in proportional asymptotic regime?

**No** our theory can also describe arbitrary spectra and large but finite $N,P,D$. In fact, the most interesting application of the theory should hold for dimension free covariates ( where $\sum_{k=1}^\infty \lambda_k < \infty$ as $D \to \infty$ ). In this case, our theoretical predictions for the risk provide an accurate approximation of the average loss rather than an exact description of the dynamics.

We added a new Appendix section F.1 which studies the deterministic equivalent for general $D,P,N$ (all large but finite). We now use this formulation in the scaling law part of the theory. We also describe this in more detail before Result 7.

#### Limitations of Current Architectures and Ablations

Some reviewers pointed out that additional experiments on more realistic architectures (incluidng Transformers with nonlinear MLP layers) would be useful.

In response to this, we are running additional experiments varying the number of attention heads, adding or subtracting MLP blocks, and examining softmax vs linear attention. We have added some initial new experiments in Figure 10.

#### Issues with writing and clarity

We have tried improving our mathematical notation and clarity throughout based on the feedback from the reviewers. With the additional page we have also expanded some of the explanation of certain results.

For example, we have

1. Simplified result 1 and included the label noise term in that expression.
2. Provided more text explaining the result in Section 4 and how it doesn't require a proportional scaling limit.
3. Added more motivation for the analysis in Appendix A in the new subsections titled **Intuitive Motivation for this Section** and **Dynamical Mean Field Theory Illustrative Example**.
4. Discussed in more detail the connection between general deep models and the looped setting we analyze mathematically for RRS covariates.
5. Tried to make mathematical notation more consistent ($\left< \right>$ for averages, reserving the symbol $i$ for $\sqrt{-1}$ instead of an index, etc.).

---

### Meta-Review · Area_Chair_CTgF · 2026-01-03

**Summary:**

The reviewers have concerns regarding the setting of tied weights, non-asymptotic analysis and more real-world experiments. The rebuttal and the new version provided substantial improvements in response to the reviewers' questions. Considering the novel theoretical contribution of push tractable scaling law for attention to the deep version, most of the concerns have been addressed by the rebuttal. I recommend a clear accept.

I suggest the authors add the additional theoretical results during the discussion period and improve the presentation and clarity.

**Reviewer Concerns:**

Reviewer KSDs' concerns have been addressed; the reply will raise the score.
Reviewer B4Ac's concerns have been solved, though the reviewer did not follow up on the discussion.
Reviewer qeR8's concerns have been solved, though the reviewer did not follow up on the discussion.
Reviewer NdPw asked for more clarification of the theoretical tools used, which have been addressed.

**Reviewer Scores:**

Reviewer KSDs already promised to raise the score from 6 to 8 after the rebuttal.
Reviewer B4Ac might not change his score, which is already 8.
I think Reviewer qeR8 will raise his score if she or he had been able to participate fully in the discussion.
Reviewer NdPw will keep his score, 6, still positively.

---

### Decision · Program_Chairs · 2026-01-26

Accept (Poster)